# Modelling organophosphate intoxication in C. elegans highlights nicotinic acetylcholine receptor determinants that mitigate poisoning

**Patricia G. Izquierdo[1]\*, Claude L. Charvet[2], Cedric Neveu[2], A. Christopher Green[3], John E. H. Tattersall[3], Lindy Holden-Dye[3], Vincent O'Connor[1]**

**1** Biological Sciences, Institute for Life Sciences, University of Southampton, Southampton, United Kingdom, **2** French National Institute for Agricultural Research (INRA), Infectiologie Animale et Santé Publique, Nouzilly, France, **3** Dstl, Defence Science and Technology Laboratory, Porton Down, Salisbury, Wiltshire, United Kingdom

\* P.Gonzalez@soton.ac.uk

## Abstract

Organophosphate intoxication via acetylcholinesterase inhibition executes neurotoxicity via hyper stimulation of acetylcholine receptors. Here, we use the organophosphate paraoxon-ethyl to treat *C. elegans* and use its impact on pharyngeal pumping as a bio-assay to model poisoning through these neurotoxins. This assay provides a tractable measure of acetylcholine receptor mediated contraction of body wall muscle. Investigation of the time dependence of organophosphate treatment and the genetic determinants of the drug-induced inhibition of pumping highlight mitigating modulation of the effects of paraoxon-ethyl. We identified mutants that reduce acetylcholine receptor function protect against the consequence of intoxication by organophosphates. Data suggests that reorganization of cholinergic signalling is associated with organophosphate poisoning. This reinforces the under investigated potential of using therapeutic approaches which target a modulation of nicotinic acetylcholine receptor function to treat the poisoning effects of this important class of neurotoxins.

## Introduction

Acetylcholinesterase inhibitors including carbamates, organophosphates and nerve agents remain an important class of neurotoxins. Their wide use in agriculture and their potential risk in bio-terrorism means they continue to require effective antidotes [11, 12]. Anti-cholinesterase poisoning raises the concentration of acetylcholine at the synapses beyond the physiological levels and triggers the persistent stimulation of the nicotinic and muscarinic receptors [1, 2]. Since acetylcholine regulates transmission at central, peripheral and neuromuscular synapses [3, 4], there is a wide spectrum of symptoms associated with anti-cholinesterase poisoning, known as cholinergic syndrome [1, 5]. Asphyxia is the main cause of death and is

**Data Availability Statement:** All relevant data are within the paper and its supporting Information Files.

**Funding:** This work was funded by the University of Southampton (United Kingdom), The Gerald Kerkut Charitable Trust (United Kingdom) and the Defence Science and Technology Laboratory, Porton Down, Wiltshire (United Kingdom). Support was received by the Institut National de Recherche pour l'Agriculture, l'Alimentation et l'Environnement (INRAE) to CLC and CN. The funders had no role in study design, data collection and analysis, decision to publish, or preparation of the manuscript.

**Competing interests:** The authors have declared that no competing interests exist.

produced by the uncontrolled stimulation of the nicotinic receptors at the neuromuscular junction of respiratory muscles [5]. The core of the treatment consists of the artificial ventilation of the victim and the injection of atropine, an antagonist of the muscarinic receptors [6]. It is usually supplemented with an oxime treatment to facilitate reactivation of the acetylcholinesterase and an anticonvulsant drug to minimize seizures during the initial cholinergic crisis [6–9]. However, the success of this treatment depends on factors such as the time of reaction between organophosphate and acetylcholinesterase, the dose of atropine administrated or the type of cholinesterase inhibitor intoxicating [6, 10, 11]. In this scenario, developing alternative treatment strategies remains pressing [12, 13].

Interestingly, although muted as a possibility, less work has been done on testing if the nicotinic receptors that mediate fast cholinergic synaptic transmission might prove a clinically useful route to treat intoxication. These acetylcholine-gated cation channels are formed by five subunits in homomeric or heteromeric combination [14]. Since multiple genes encode for nicotinic receptor subunits, the combinatorial complexity might provide an opportunity to selectively modulate the cholinergic signal [14]. Receptor subtypes exhibit different biophysical properties and some subtypes can be predominately expressed with respect to others depending on external stimuli [15–17]. Pharmacological activation of nicotinic receptors by exogenous agonists like nicotine is an established route to modulation of receptor composition and function [18–21]. Finally, the interaction of nicotinic receptors with auxiliary proteins modifies their trafficking, clustering, sensitivity or motility between synaptic and extrasynaptic domains with profound impacts on the overlying efficacy of cholinergic transmission [22–27]. Understanding the molecular mechanisms that regulate the cholinergic signal at all levels could be critical to develop mitigating treatments in the context of anti-cholinesterase intoxication.

Despite its simplicity, the free-living nematode *C. elegans* can exhibit drug-induced behavioural plasticity and is well suited to investigating pharmacological and genetic determinants of cholinergic signalling [28, 29]. In particular, preconditioning of nematodes to nicotine modifies the consequent phenotype of these worms when they are post-exposed to the same signal [30–33]. Alternatively, the chronic exposure to nicotine triggers the habituation of egg-laying, a cholinergic-dependent behaviour [34, 35]. The genetic amenability of the nematode combined with a well-defined set of behaviours and the characterization of its nervous system make *C. elegans* an attractive model organism to research drug related modulation including those associated with cholinergic transmission [28, 33–35].

In previous experiments, we built on the established work of others and demonstrated the potential of the organism model *C. elegans* to investigate acetylcholinesterase intoxication and recovery [36]. Specifically, we highlighted the measurement of pumping rate on food as a suitable cholinergic-mediated behaviour that shows a time and dose dependent inhibition by the presence of anti-cholinesterases. This inhibition can be restored when nematodes are removed from the drug [36]. Pharyngeal pumping under these conditions directly reflects cholinergic transmission at the body neuromuscular junction that leads to a sustained contraction and shrinkage of the worm [37]. Here, we demonstrate that nematodes exhibit a cholinergic modulation in two different paradigms, the precondition to low doses of the drug and the chronic stimulation with high intoxicating concentrations. The preconditioning paradigm with the organophosphate paraoxon-ethyl intensified the behavioural effect of the drug when nematodes are post-exposed. However, the incubation with high concentrations of paraoxon-ethyl triggers, a within drug, mitigating recovery of the cholinergic-dependent pharyngeal pumping. We identified mutants that impact the synaptic organization and/or sensitivity of the nicotinic receptors at the neuromuscular junction as important determinants of this within drug mitigation of poisoning. This finding provokes the notion that modulating the tone of the nicotinic

acetylcholine receptor function during organophosphate intoxication is an underexplored route to mitigate organophosphate poisoning.

## Results

### The pharyngeal microcircuit of *C. elegans* exhibits mitigating aldicarb-induced plasticity after preconditioning with sub-maximal dose of the drug

As a paradigm of behavioural plasticity in *C. elegans*, we investigated how the preconditioning to acetylcholinesterase inhibitors impacted on the subsequent response to drug treatment. This was done by quantifying pumping phenotype on food, a cholinergic-dependent behaviour that exhibits a dose-time dependent inhibition in response to the exposure of anti-cholinesterase [36]. We observed that nematodes exposed to 50 μM aldicarb for 24 hours exhibited a 50% drug-dependent inhibition of the pharyngeal function. However, recovery from inhibition was observed 2 hours after being removed from the drug (Fig 1A).

Based on the above, nematodes were incubated either on control plates or 50 μM aldicarb-containing plates for 24 hours and then transferred on non-drugged plates for 2 hours to allow the recovery of the pharyngeal function. Finally, control and aldicarb-treated worms were intoxicated on plates containing fivefold increase of anti-cholinesterase concentration where pump rate on food was measured. (Fig 1B). Wild type nematodes exposed to the maximal concentration of aldicarb exhibited a time-dependent inhibition of the feeding phenotype, being completely abolished after 24 hours incubation [36]. Interestingly, the pharyngeal pumping rate of nematodes pre-exposed to the cholinesterase inhibitor aldicarb was less susceptible to inhibition than non-preconditioned worms when they were subsequently intoxicated with the maximal dose. This was clearly evidenced at 6 and 24 hours after being transferred to the 250 μM aldicarb-containing plates (Fig 1C).

The result indicates that the preconditioning step with the carbamate aldicarb, induces a reduced drug sensitivity in the pharyngeal phenotype of pre-exposed worms when they are subsequently exposed to a maximal drug concentration.

### Preconditioning with sub-maximal dose of paraoxon-ethyl leads to an enhanced drug-dependent inhibition of pharyngeal pumping

Carbamates and organophosphates are two distinct groups of cholinesterase inhibitors that cause similar cholinergic toxicity [2]. In order to investigate the behavioural modulation of pharyngeal pumping that emerges from organophosphate preconditioning, we performed an equivalent experiment to that described above with the organophosphate paraoxon-ethyl (Fig 2A). We previously demonstrated that paraoxon-ethyl inhibits *C. elegans* acetylcholinesterases in a reversible manner [36]. This triggers an inhibition of the pharyngeal function that is recoverable when nematodes are removed from drugged plates [36]. In the present study, wild type nematodes were preconditioned for 24 hours on plates containing 20 μM paraoxon-ethyl. Similar to the aldicarb experiment, this concentration and time of exposure was selected to reduce the pumping phenotype by half of the maximal response after 24 hours of intoxication. This inhibition was recovered 3 hours after being removed from the drugged plates [36]. Finally, the preconditioned and non-preconditioned worms were transferred to 100 μM paraoxon-ethyl, a fivefold higher concentration than used in the preceding preconditioning step (Fig 2A).

In contrast to the results observed for aldicarb, the pre-exposure with paraoxon-ethyl caused an enhanced inhibition of pumping compared to the control non-precondition treatment. This was evidenced by the residual pumping observed 3 hours after transferring

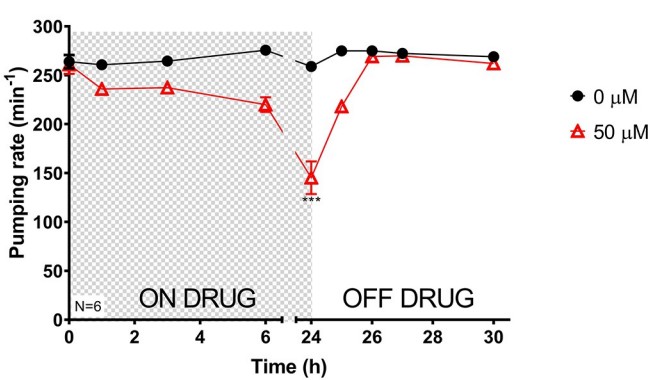

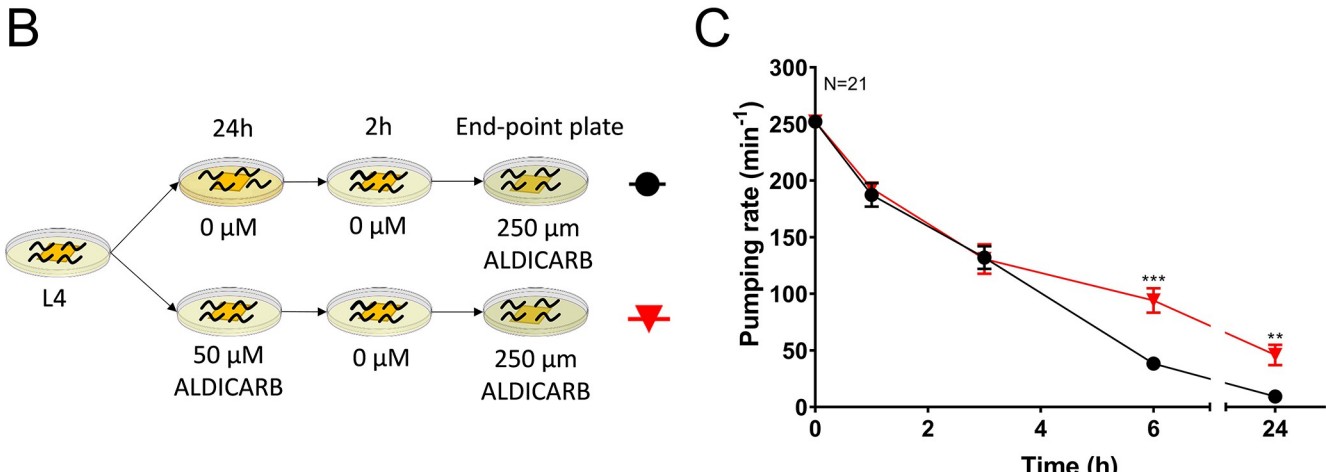

**Fig 1. Aldicarb preconditioning reduces drug-induced inhibition of pharyngeal pumping.** A) Pharyngeal pumping rate was quantified at indicated times for synchronized nematodes exposed to a sub-lethal dose of aldicarb. Nematodes were transferred to non-drug containing plates where the recovery of the pharyngeal function was observed at the indicated times. The shaded box indicates period of treatment. Data are shown as mean ± SEM of 6 worms in 3 independent experiments (N = 6). Statistical significance between exposed and non-exposed nematodes was calculated by two-way ANOVA test followed by Bonferroni correction. **p≤0.01. B) Synchronized L4 worms were incubated on 50 μM aldicarb plates. Non-exposed nematodes were used as controls. After 24 hours, worms were transferred to non-drug plates to allow the recovery of the pharyngeal function before subsequent exposure to 250 μM aldicarb for measurement of pharyngeal pumping C) Preconditioned worms exhibited higher pharyngeal pumping rate than non-preconditioned animals 6 and 24 hours after being transferred to plates containing 250 μM aldicarb. Data are mean ± SEM of 21 worms in at least 11 independent experiments (N = 21). Statistical significance between preconditioned and non-preconditioned nematodes was calculated by two-way ANOVA test followed by Bonferroni correction. **p≤0.01; ***p≤0.001.

nematodes to the maximal drug concentration plates (Fig 2B). This increased inhibition could be explained by either an increased sensitivity to paraoxon-ethyl or an incomplete recovery of the acetylcholinesterase inhibition imposed by the initial exposure phase of the protocol. Recovery studies from other organophosphate intoxication demonstrate that acetylcholinesterase activity was incompletely recovered even when nematodes present a normal phenotype based on visual observations [38, 39]. In order to investigate this, we supplemented the recovery plate with 2 mM obidoxime (Fig 2C). We previously demonstrated that obidoxime improves the acetylcholinesterase activity and the pharyngeal function of *C. elegans* during the recovery from paraoxon-ethyl inhibition [36]. Similar to previous observations,

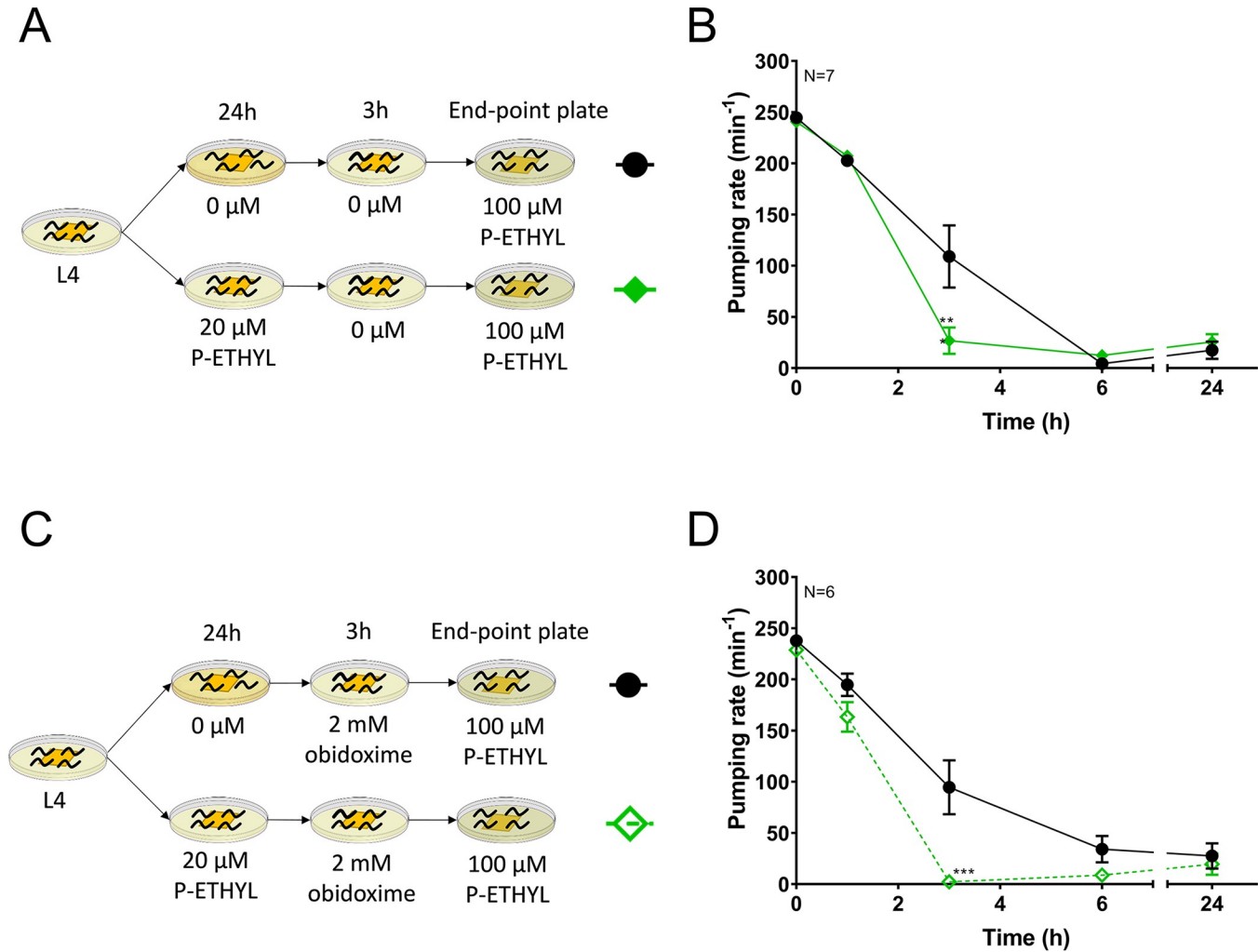

**Fig 2. Nematodes preconditioned with paraoxon-ethyl are sensitized to subsequent organophosphate inhibition of pharyngeal pumping.** A) Synchronized L4 worms were incubated on either non-drug or 20 μM paraoxon-ethyl containing plates. After 24 hours, they were transferred to non-drug containing plates to allow the recovery of pharyngeal function. They were transferred to plates containing 100 μM paraoxon-ethyl where the pharyngeal pumping was scored. B) Paraoxon-ethyl-preconditioned nematodes exhibited a greater reduction of the pharyngeal pumping following transfer to 100 μM paraoxon-ethyl plates compared to the non-preconditioned animals. Data are shown as mean ± SEM of 7 worms in at least 4 independent experiments (N = 7). C) Nematodes were preconditioned as indicated in A, however, obidoxime was added to the recovery plate to promote the rescue of the acetylcholinesterase activity after paraoxon-ethyl inhibition. D) Preconditioned nematodes exhibited a similar response to maximal dose of paraoxon-ethyl when they were allowed to recover in the presence or in the absence of obidoxime. Data are shown as mean ± SEM of 6 worms in at least 3 independent experiments (N = 6). Statistical significance between preconditioned and non-preconditioned nematodes was calculated by two-way ANOVA test followed by Bonferroni corrections. ***$p \leq 0.001$.

preconditioned nematodes remained more susceptible to the maximal dose of paraoxon-ethyl 3 hours after incubation compared to those never exposed to the drug (Fig 2D). This indicates that the enhanced behavioural change observed in the nematodes preconditioned with paraoxon-ethyl is due to an increased sensitivity to the drug rather than a residual inhibition of the worm acetylcholinesterase.

The distinct pattern of drug effects on pharyngeal behaviour observed between aldicarb and paraoxon-ethyl preconditioned worms suggests differences in the drug-dependent adaptation to the acetylcholinesterase inhibition by carbamates and organophosphates despite them sharing a core mode of action. This might include contributions from non-overlapping targets for the two classes of distinct acetylcholinesterase inhibitors.

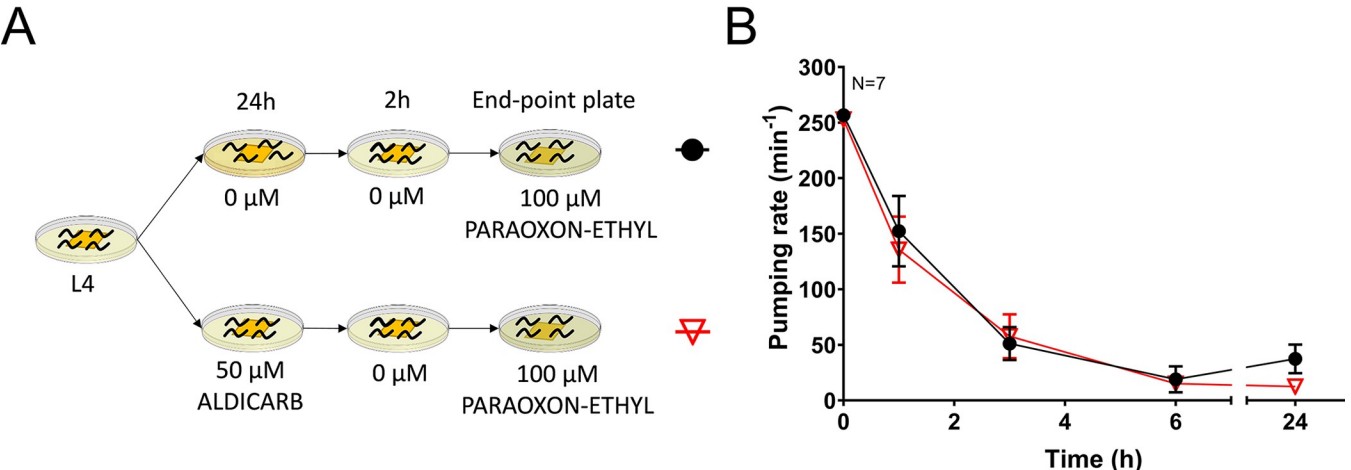

**Fig 3. Aldicarb-preconditioned and non-preconditioned nematodes exhibit a similar pharyngeal function when subsequently exposed to paraoxon-ethyl.**
A) Synchronized L4 worms were incubated on either vehicle control or 50 μM aldicarb plates for 24 hours. After recovery on non-drug containing plates, they were exposed to 100 μM paraoxon-ethyl and pumping was measured. B) Aldicarb pre-exposed nematodes exhibit a similar sensitivity to paraoxon-ethyl as non-preconditioned worms. Data are shown as mean ± SEM of 7 worms in at least 4 independent experiments (N = 7).

## Aldicarb does not precondition a change in the pharyngeal response to paraoxon-ethyl

In order to interrogate the nature of the distinct preconditioning outcomes with carbamates and organophosphates, nematodes were pre-exposed with aldicarb and then post-exposed to paraoxon-ethyl (Fig 3A). Non-preconditioned nematodes and those preconditioned with 50 μM aldicarb for 24 hours exhibited a similar inhibition pattern of the pumping rate when they were subsequently exposed to 100 μM paraoxon-ethyl (Fig 3B).

Overall, the preconditioning with aldicarb provoked the adaptation of the pharyngeal pumping to post-exposure with aldicarb but not to the post-exposure with paraoxon-ethyl. This reinforced our hypothesis that nematodes are able to adapt to anti-cholinesterase drug exposure by the previous experience. However, this is achieved by distinct mechanisms for carbamates and organophosphates. This clearly indicates that carbamates are not a well-suited model to detail investigation of the processes that underlie organophosphate intoxication, recovery and drug related plasticity, even though they both act via inhibition of acetylcholinesterase [2].

## *C. elegans* adults exhibit spontaneous recovery of the NMJ function in the presence of higher doses of paraoxon-ethyl

In addition, to investigating the impact of preconditioning, we studied if chronic exposure of *C. elegans* impacted the pharyngeal pumping within the drug dosing regimen [29]. In the context of organophosphate poisoning, this implies regulation during exposure that causes a high level of acetylcholinesterase inhibition and pronounced hyperstimulation of the cholinergic receptors in the postsynaptic terminal [36]. In order to investigate how the cholinergic system of *C. elegans* responds to inhibition with high concentrations of organophosphates, we quantified the pumping rate during the sustained exposure of wild type nematodes to 500 μM paraoxon-ethyl. The concentration was calculated as 25-fold higher than the IC50 value for the drug-induced inhibition pharyngeal phenotype at 24 hours. This would model lethal doses that represent highly toxic environmental exposure [36]. This overstimulation initially caused the complete inhibition of pharyngeal pumping after 3 hours incubation in the drug (Fig 4A).

Remarkably, after this first inhibition, despite the sustained exposure to paraoxon-ethyl, nematodes exhibited a spontaneous recovery of the pumping rate at 6 hours. This recovery of the feeding phenotype was not sustained, and the pump rate was subsequently abolished at the 24 hours of exposure to paraoxon-ethyl (Fig 4A).

The effect of the sustained exposure to 500 μM paraoxon-ethyl was also investigated in a different cholinergic-dependent phenotype at the level of the body wall neuromuscular junction by measuring body length. As expected, the intoxication with paraoxon-ethyl caused hypercontraction of the body wall muscles, evidenced by the shrinkage of nematodes. Wild type worms poisoned with 500 μM paraoxon-ethyl exhibited a 50% reduction of the body length compared to non-exposed nematodes after 1 hour on the drug containing plates (Fig 4B). We previously demonstrated that this reduction is the maximum level of shrinkage nematodes can reach when they are incubated to anti-cholinesterases [36]. Similar to the pharyngeal function, the body length was partially recovered at 6 hours of incubation despite the continued presence of the drug. This recovery was transient and reversed, as the maximum shrinkage was again observed at 24 hours of exposure (Fig 4B).

Overall, the results indicate that the overstimulation of the cholinergic pathways by high doses of paraoxon-ethyl triggers a pharyngeal and body wall behavioural response characterized by three phases: an initial inhibition, a partial and transitory recovery in face of ongoing drug inhibition followed by a subsequent inhibition. This highlights an adaptation of the behaviours tested even in the continuous presence of paraoxon-ethyl. Furthermore, the similar recovery of cholinergic function for pumping rate and body length when nematodes are continuously exposed to the drug suggests a common underpinning mechanism.

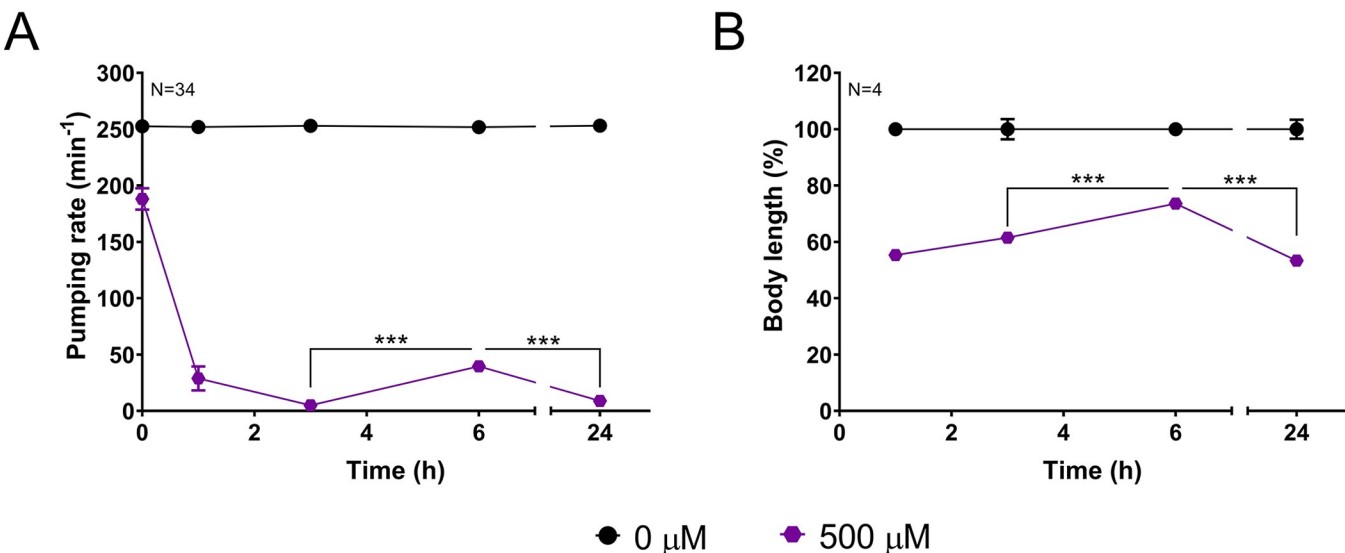

**Fig 4. Pharyngeal and body wall neuromuscular behaviours exhibit a paraoxon-ethyl intoxication pattern characterized by three phases, an initial inhibition, a spontaneous in drug recovery and a subsequent inhibition.** A) Pharyngeal pumping in nematodes exposed to 500 μM paraoxon-ethyl display a complete inhibition of pumping at 3 hours and an in-drug recovery at 6 hours. The complete inhibition of pumping is observed after 24 hours of exposure. Data are shown as mean ± SEM of 34 worms in at least 17 independent experiments (N = 34). B) Nematodes were exposed to 500 μM paraoxon-ethyl and body length recorded 1, 3, 6 and 24 hours of exposure. Percentage of body length was referenced against the corresponding age-matched untreated control. Similar to pumping the three phases of intoxication were observed encompassing: initial shrinkage of nematodes at 1 hour, spontaneous body length recovery after 6 hours and the subsequent shrinkage of length at 24 hours of exposure. Data are shown as mean ± SEM of 6 worms in 4 independent experiments (N = 6). Statistical significance was calculated by two-way ANOVA test followed by Bonferroni corrections. **p≤0.01; ***p≤0.001.

### The molecular determinants of the pharyngeal neuromuscular junction are not involved in the cholinergic plasticity observed in the pharynx

Uncovering the signalling pathways that underpin the capacity of cholinergic-dependent behaviours to exhibit mitigating in-drug recovery might suggest additional targets that palliate aspects of organophosphate intoxication. In order to identify molecular components of this cholinergic modulation, we compared the pumping rate of different mutant worms with the wild type control in the presence or absence of 500 μM paraoxon-ethyl at 3, 6 and 24 hours of exposure. We defined these time points as key intervals in the experiment to detect initial inhibition, spontaneous recovery and subsequent inhibition. The screening utilized the pharyngeal pumping as bio-assay since we previously demonstrated the potential of this phenotype in the research of organophosphate intoxication and recovery [36].

We first screened the pumping rate of strains containing mutations in important components of the pharyngeal neuromuscular junction (Fig 5). This included the acetylcholinesterase ACE-3 [40, 41], the nicotinic receptor subunit EAT-2 [42, 43], the muscarinic receptor GAR-3 [44] and the glutamate-gated chloride channel subunit AVR-15 [45]. All these proteins are critical to the contraction-relaxation cycle of the pharyngeal muscles responsible for pumping. All mutants tested, except *eat-2 (ad465)*, exhibited the three phases encompassing an initial inhibition of pumping, rebound recovery and reoccurring inhibition similar to the wild type control exposed to paraoxon-ethyl (Fig 5). This indicates that these proteins are not key determinants of the drug-induced plasticity in the pharyngeal pumping.

The pumping rate of *eat-2 (ad465)* deficient worms exposed to paraoxon-ethyl was continuously inhibited over the time and lacked the spontaneous within drug recovery observed in

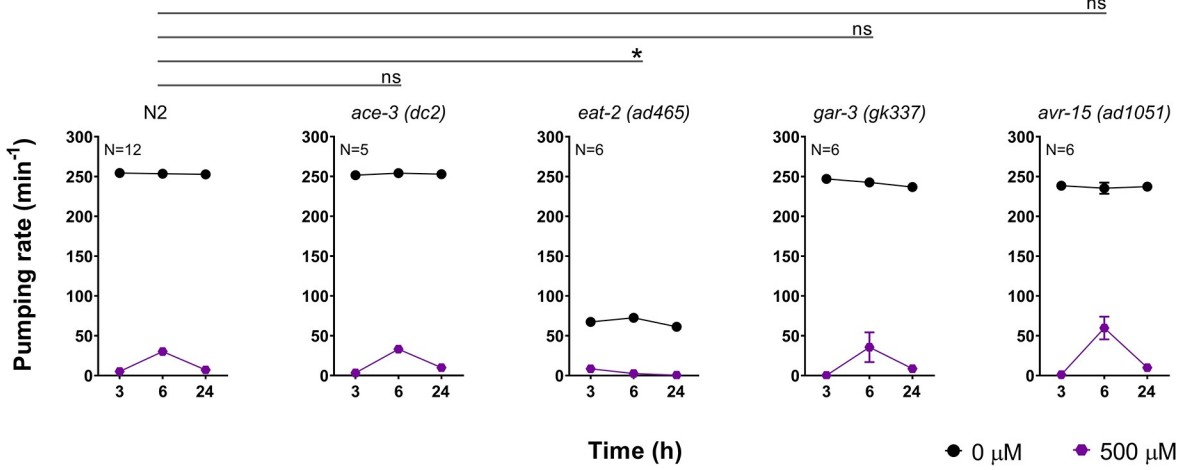

**Fig 5. The paraoxon-ethyl induced changes in pharyngeal pumping are not elicited by components underpinning pharyngeal neuromuscular function.** N2 wild type nematodes continuously exposed to 500 μM paraoxon-ethyl exhibited in drug recovery of the pharyngeal function at 6 hours followed by a subsequent inhibition at 24 hours. Data are shown as mean ± SEM of 16 worms in at least 8 independent experiments (N = 16). The acetylcholinesterase ACE-3 of *C. elegans* is specifically expressed in the isthmus of the pharynx [41]. ACE-3 deficient nematodes exposed to 500 μM paraoxon-ethyl exhibit a similar in-drug recovery followed by inhibition of the pharyngeal pumping rate compared to wild-type worms. Data are shown as mean ± SEM of 5 worms in at least 3 independent experiments (N = 5). Nematodes lacking EAT-2 did not exhibit the in drug recovery observed in wild type worms. Data are shown as mean ± SEM of 6 worms in at least 3 independent experiments (N = 6). The muscarinic receptor GAR-3 is expressed in the isthmus and is involved in the feeding movement [46]. Mutant nematodes lacking GAR-3 exhibited a similar paraoxon-induced plasticity of the pumping rate compared to the wild type worms. Data are shown as mean ± SEM of 6 worms in at least 3 independent experiments (N = 6). E) *avr-15* encodes a glutamate-gated chloride channel subunit responsible for the relaxation of the pharyngeal muscle upon contraction [45]. *avr-15* mutant worms exhibit a similar pattern of pharyngeal pumping rate than wild type animals intoxicated on 500 μM paraoxon-ethyl plates. Data are shown as mean ± SEM of 6 worms in at least 3 independent experiments (N = 6). Statistical significance was calculated by two-way ANOVA test followed by Bonferroni corrections. $^{ns}p > 0.05$; $^*p \leq 0.05$.

wild type nematodes at 6 hours (Fig 5). This might hint at a role for EAT-2 in the paraoxon-induced plasticity. However, this view should be tempered by the intrinsic blunting of food-induced pumping in these mutant worms.

Overall, these results indicate that the within drug recovery observed in the pharyngeal function of nematodes exposed to paraoxon-ethyl is not determined by molecular components of the pharyngeal neuromuscular junction.

## Body wall muscle receptor function controls drug-induced pumping inhibition and within drug recovery

We previously demonstrated that pharmacological activation of the body wall neuromuscular junction by either aldicarb or levamisole exerts an indirect inhibition of the pharyngeal pumping [37]. The chaperone RIC-3 and the L-type receptor subunits UNC-29 and LEV-1 are key determinants of this response and therefore are known as pharmacological determinants of the pharyngeal function [37].

Accordingly, we investigated if these determinants could be involved in the paraoxon-induced plasticity observed in the pharyngeal pumping of wild type worms. The pumping rate of *ric-3*, *unc-29* and *lev-1* mutant nematodes was measured in the presence or absence of 500 μM paraoxon-ethyl at 3, 6 and 24 hours and the results were compared with the paired wild type control.

We observed that nematodes deficient in the ancillary protein RIC-3 and the non-alpha LEV-1 subunit of the L-type receptor exhibited a strong resistance to the pharyngeal inhibition by 500 μM paraoxon-ethyl (Fig 6). This is consistent with our previous observations using the cholinesterase inhibitor aldicarb [37]. In addition, these mutants did not show an obvious drug-induced plasticity of the pumping rate at this concentration (Fig 6). However, the intrinsic resistance to inhibition of the pharyngeal pumping by paraoxon-ethyl experienced by these two strains could preclude the observation or the actual expression of the drug-induced plasticity characteristic of the N2 wild type. Accordingly, we exposed nematodes deficient in *ric-3 (hm9)* or *lev-1 (e211)* to a higher dose of the cholinesterase inhibitor. The pumping rate of both strains intoxicated with 1 mM paraoxon-ethyl exhibited the initial inhibition and the spontaneous recovery phases mimicking the wild type response. However, the subsequent inhibition of the pharyngeal function observed in wild type animals is absence in *ric-3 (hm9)* and *lev-1 (e211)* strains (Fig 6).

Interestingly, nematodes deficient in the other non-alpha UNC-29 subunit of the L-type receptor were not resistant to the pharyngeal inhibition by paraoxon-ethyl (Fig 6). However, this strain presented a similar pronounced and sustained spontaneous recovery of the pharyngeal function as *lev-1 (e211)* and *ric-3 (hm9)* nematodes in the presence of 500 μM and 1 mM of paraoxon-ethyl. The three mutant strains, *ric-3 (hm9)*, *lev-1 (e211)* and *unc-29 (e193)*, lacked the subsequent inhibition of pumping that follows the spontaneous recovery observed in the wild type worms (Fig 6). These data show that the sustained within drug recovery is not a peculiarity of mutants that are insensitive to pharyngeal inhibition by paraoxon-ethyl, indicating a dissociation between the determinants of drug sensitivity and within drug-induced recovery. While RIC-3 and LEV-1 are involved in both processes, UNC-29 is only involved in the expression of the subsequent inhibition that occurs after the spontaneous recovery observed in wild type worms exposed to 500 μM paraoxon-ethyl.

In order to address the cellular determinants of this, we performed the paraoxon-ethyl intoxication experiment with transgenic expression of *lev-1* and *unc-29* in the body wall muscle of *lev-1 (e211)* and *unc-29 (e193)* mutant background, respectively (Fig 7). The expression of GFP in the coelomocytes of these two mutant backgrounds was used as co-injection marker

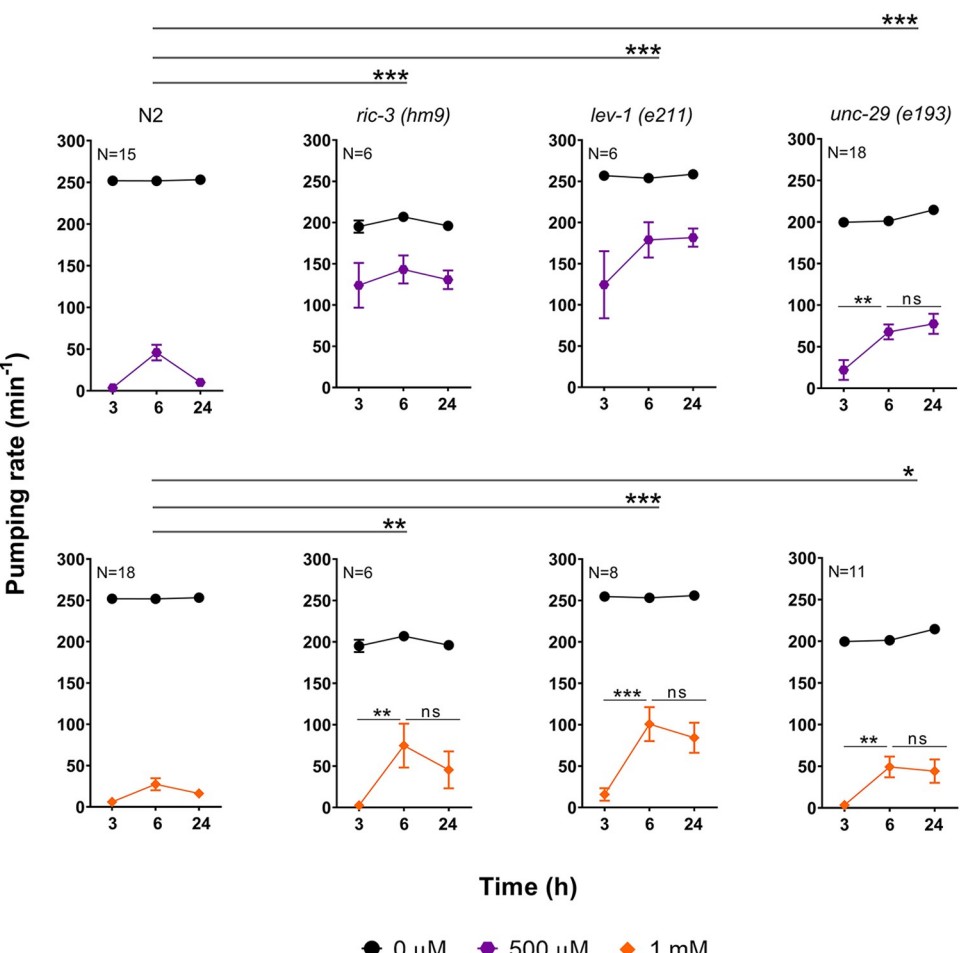

**Fig 6. Nematodes deficient in the non-alpha subunits of the L-type body wall muscle receptor, LEV-1 and UNC-29, exhibited a sustained in drug recovery of the pharyngeal pumping.** Paraoxon-ethyl induced plasticity of the pharyngeal function in wild type nematodes exposed to 500 μM and 1 mM. Data are shown as mean ± SEM of 15 worms in 8 independent experiments (N = 15) or 18 worms in 9 independent experiments (N = 18), respectively. Nematodes deficient in the chaperone protein RIC-3 exhibited resistance to the inhibition of the pumping in the presence of 500 μM of paraoxon-ethyl. The exposure to 1 mM concentration inhibited the pumping rate after 3 hours and showed an in drug spontaneous recovery sustained for up to 24 hours. Data are shown as mean ± SEM of 6 worms in 3 independent experiments for each concentration (N = 6). *lev-1* encodes a non-alpha subunit of the L-type receptor. LEV-1 lacking nematodes phenocopy the paraoxon-ethyl induced sustained in drug recovery of *ric-3* deficient nematodes. Data are shown as mean ± SEM of 6 worms in 3 independent experiments (N = 6) for 500 μM exposure or 8 worms in 4 independent experiments for 1 mM exposure. UNC-29 is the other non-alpha subunit of the L-type receptor. Nematodes deficient in UNC-29 exhibited wild type resistance to paraoxon-ethyl but a sustained in drug recovery of the pharyngeal pharyngeal pumping after 24 hours in 500 μM and 1 mM. Data are shown as mean ± SEM of 18 worms in 9 independent experiments (N = 18) or 11 worms in 6 independent experiments (N = 11), respectively. Statistical significance was calculated by two-way ANOVA test followed by Bonferroni corrections. $^{ns}p>0.05$; $^*p\leq0.05$; $^{**}p\leq0.01$; $^{***}p\leq0.001$.

in the body wall rescue lines. Although this expression in *lev-1 (e211)* slightly modify the pharyngeal inhibition pattern in the presence of the drug, the introduction of the wild type version of *lev-1* and *unc-29* in their respective mutant strain rescued the wild type pharyngeal sensitivity to paraoxon-ethyl. This reinforces our previous data indicating that LEV-1 and UNC-29 are both pharmacological determinants of the drug-induced inhibition of pharyngeal function [37]. However, while the introduction of *lev-1* into CB211 strain rescued the three phases

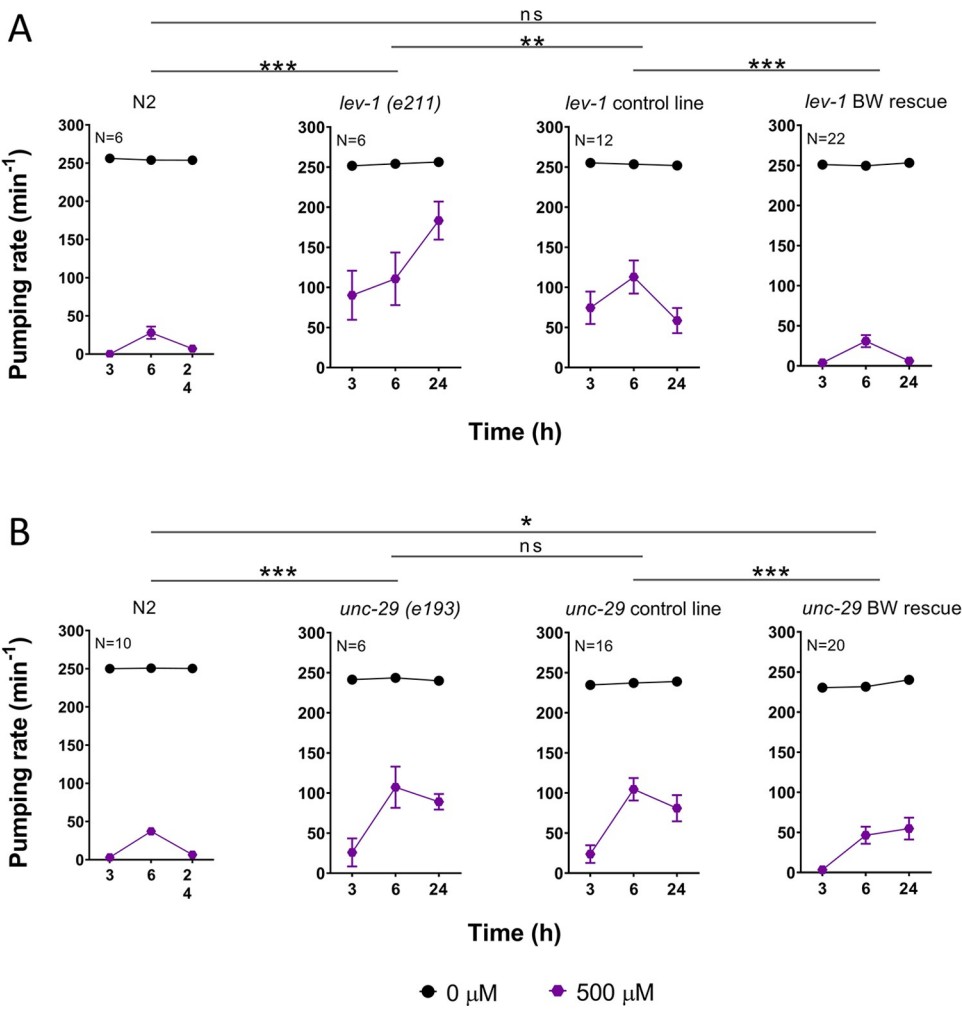

**Fig 7. Body wall muscle rescue of the non-alpha subunits LEV-1 and UNC-29 restores wild type sensitivity to prolonged paraoxon-ethyl exposure.** A) The pharyngeal sensitivity and the three phases characteristic of drug-induced plasticity to paraoxon-ethyl in the *lev-1* deficient worms were restored by introducing the wild type version of the gene selectively in the body wall muscles under control of the *myo-3* promoter. Data are shown as mean ± SEM of 6 worms in 3 independent experiments for N2 and *lev-1 (e211)* strains (N = 6); 12 worms in 6 independent experiments of 2 independent lines for *lev-1* control line (N = 12) and 22 worms in 11 independent experiments of 4 independent lines for *lev-1* BW rescue (N = 22). The genotype of control and BW rescue lines corresponds to CB211 *lev-1 (e211) IV*; Ex[P*unc-122::gfp*] and CB211 *lev-1 (e211) IV*; Ex[P*unc-122::gfp*; P*myo-3::lev-1*], respectively. B) The introduction of the wild type UNC-29 in the body wall muscles of *unc-29* mutant worms rescued the consequent inhibition of the pharyngeal function that follows the spontaneous recovery in paraoxon-ethyl exposed worms. Data are shown as mean ± SEM of 10 worms in 5 independent experiments for N2 (N = 10); 6 worms in 3 independent experiments for *unc-29 (e193)* (N = 6); 16 worms in 8 independent experiments of 3 independent lines for *unc-29* control line (N = 16) and 20 worms in 10 independent experiments of 3 independent lines for *unc-29* BW rescue (N = 20). The genotype of control and BW rescue lines corresponds to CB193 *unc-29 (e193) I*; Ex[P*unc-122::gfp*] and CB193 *unc-29 (e193) I*; Ex[P*unc-122::gfp*; P*myo-3::unc-29*], respectively. Statistical significance was calculated by two-way ANOVA test followed by Bonferroni corrections. $^{ns}p > 0.05$; $^*p \leq 0.05$; $^{**}p \leq 0.01$; $^{***}p \leq 0.001$.

distinctive of the organophosphate-induced plasticity observed in the pharyngeal phenotype of wild type worms (Fig 7A), the introduction of *unc-29* into CB193 strain did not (Fig 7B). This highlights the importance of the LEV-1 subunit, and therefore the body wall L-type receptor, in the subsequent inhibition of the pharyngeal function that occurs after the spontaneous recovery in the presence of paraoxon-ethyl.

## The sensitivity of the L-type receptor at the body wall muscle is responsible for the pumping inhibition that follows the spontaneous recovery in the presence of paraoxon-ethyl

LEV-1 and UNC-29 are the two non-alpha subunits that combine with the three alpha subunits UNC-63, UNC-38 and LEV-8 to compose the L-type receptor at the body wall muscles of nematodes [47]. However, the non-alpha subunits of these receptors create the binding pocket for neurotransmitter only when they are combined with other alpha subunit [48]. To investigate the specificity of these two subunits in the paraoxon-ethyl plasticity response, we analysed the behaviour of different mutant strains in these genes in the presence of 500 μM paraoxon-ethyl (Fig 8). The *x427* allele of *lev-1* consists of 1,267 bp deletion that contains exon 4. This results in a LEV-1 protein that lacks the first, second and third transmembrane domains (S1 Fig). The *e211* mutation consists of a missense substitution of glycine to glutamate in the fourth transmembrane domain of LEV-1 (S1 Fig). Interestingly, nematodes with the *x427* allele of *lev-1* exhibited wild type sensitivity to paraoxon-ethyl and do not phenocopy the pharyngeal response in the presence of the drug characteristic of the strain harbouring the *e211* mutation (Figs 6 and 8).

The strain CB1072 is considered a null mutant strain of *unc-29* [53]. The *e1072* allele contains a G to A base substitution in the splicing acceptor site of intron 8 (S2 Fig). This causes a new splice acceptor that utilizes the first G in exon 9 creating a frameshift mutation and a premature stop codon between the first and the second transmembrane domain of the protein (S2 Fig). However, the *e193* allele contains a missense mutation that substitutes a conserved proline for a serine in the loop connecting the second and third transmembrane domain (S2 Fig). Similar to *lev-1* deficient strains, the null mutant allele of *unc-29 (e1072)* did not exhibit the

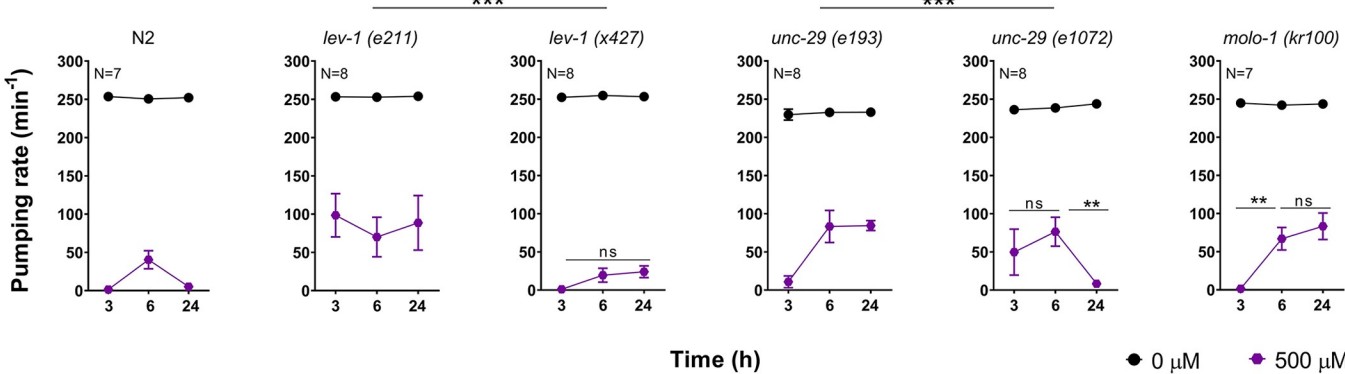

**Fig 8. The efficacy of the L-type receptor is a significant determinant of the spontaneous recovery of pharyngeal pumping in nematodes exposed to paraoxon-ethyl.** Paraoxon-induced pharyngeal plasticity in wild type nematodes incubated with 500 μM organophosphate. Data are shown as mean ± SEM of 14 worms in at least 7 independent experiments (N = 14). *lev-1 (e211)* mutant strain contains a single point mutation in the fourth transmembrane domain. This confers resistance to paraoxon-ethyl induced inhibition of the pharyngeal pumping. Data are shown as mean ± SEM of 8 worms in at least 4 independent experiments (N = 8). The mutation of *lev-1 (x427)* strain consists of a rearrangement of the genomic sequence that prevents the transcription of the gene into a protein [49]. This is the reference null-mutant of LEV-1. This mutation does not phenocopy the pharyngeal response to paraoxon-ethyl observed in *lev-1 (e211)* mutant background. Data are mean ± SEM of 8 worms in at least 4 independent experiments (N = 8). The mutation of *unc-29 (e193)* strain consists of a single point proline to serine in the loop connecting the second and third transmembrane domain of the subunit. This proline is highly conserved in the cys-loop receptor subunits and is implicated with the gating of the receptor [50–52]. This mutation conferred a sustained paraoxon-ethyl induced plasticity. Data are shown as mean ± SEM of 8 worms in at least 4 independent experiments (N = 8). In contrast, the null-mutant *unc-29 (e1072)* exhibited resistance to the pharyngeal inhibition by paraoxon-ethyl but did not express paraoxon-ethyl induced plasticity. Data are shown as mean ± SEM of 8 worms in at least 4 independent experiments (N = 8). MOLO-1 is an auxiliary protein implicated in the positive modulation of the L-type receptor [24]. *molo-1* lacking worms exposed to paraoxon-ethyl exhibited spontaneous recovery of the pharyngeal function that was sustained compared to wild type worms. Data are shown as mean ± SEM of 7 worms in at least 4 independent experiments (N = 7). Statistical significance was calculated by two-way ANOVA test followed by Bonferroni corrections. ***p≤0.001.

sustained recovery of the pharyngeal function in the presence of paraoxon-ethyl characteristic of the *e193* allele (Fig 8).

If we hypothesise that *lev-1 (e211)* and *unc-29 (e193)* genes harbouring non-null mutations encode subunits that could be incorporated into the mature L-type receptor, our data suggest that the resulting receptor might express an assembled receptors with altered function. This modification could be involved in the sustained recovery of the pharyngeal function after initial paraoxon-induced inhibition occurred. The fact that the paraoxon-induced plasticity in the pharyngeal phenotype observed in *lev-1 (e211)* and *unc-29 (e193)* mutant strains phenocopy the paraoxon response observed in the mutant strain *molo-1 (kr100)* supports the hypothesis (Fig 8). Since MOLO-1 is an auxiliary protein that acts as positive modulator of the L-type receptor function [24], the data support the notion that the *lev-1 (e211)* and *unc-29 (e193)* mutations might alter receptor function.

To investigate this hypothesis, the L-type acetylcholine receptor of *C. elegans* was expressed in *Xenopus* oocytes by co-injecting cRNAs for the five subunits that co-assemble to generate the ion channel (UNC-63, UNC-38, UNC-29, LEV-1 and LEV-8) along with the three ancillary proteins (UNC-50, RIC-3 and UNC-74) as previously reported [47]. The wild type version of either LEV-1 or UNC-29 was replaced by the mutated version corresponding to the mutations *e211* and *e193*, respectively, and the response to 300 µM acetylcholine was used to compare the amplitude of the currents generated by the different populations of receptors (Fig 9). The co-expression of either *lev-1 (e211)* or *unc-29 (e193)* mutant subunits along with the wild type cRNAs of the other components of the L-type receptor in oocytes, significantly reduced the current amplitude of the resulting receptor (Fig 8). However, omission of any of the eight genes required to constitute the ion channel fails to generate functional channels when expressed in the *Xenopus* oocyte [47]. Thus, our observation supports the hypothesis that the mutated subunit assembles in the functional receptor but disrupt the response to acetylcholine in the resulting ion channel. In addition, the current recorded from oocytes that co-express both the mutated and the wild type version of *lev-1* and *unc-29* in a 1:1 ratio, respectively, is significantly larger than the current measured from oocytes that contained the mutated version of either of these two genes alone. However, these currents generated from receptors assembled from competing wild type and mutant versions of *lev-1* or *unc-29* were significantly reduced compared to currents made from receptors exclusively made of wild type version of the L-type receptor components. This is consistent with the wild type and the mutated version of the protein competing for the formation of the mature ion channel, resulting in a dominant-negative effect on the L-type receptor function.

Overall, the data indicate that *lev-1 (e211)* and *unc-29 (e193)* encode for subunits that can be inserted into the mature receptor in a way that appears to disrupt the normal function of the co-assembled receptor.

## The location of the L-type receptor at the body wall neuromuscular junction might be involved in the spontaneous recovery of the pharyngeal pumping in the presence of paraoxon-ethyl

Prompted by the important modulatory role of the L-type receptor in allowing the expression of mitigating plasticity to paraoxon-ethyl exposure, we tested the pharyngeal pumping to paraoxon-ethyl intoxication in mutants deficient in different auxiliary proteins of this receptor's function. RSU-1 and OIG-4 are neuromuscular junction proteins that play an important role in organizing the L-type receptor within the body wall neuromuscular junction [54, 55]. These mutants exhibited wild type sensitivity to paraoxon-ethyl exposure (Fig

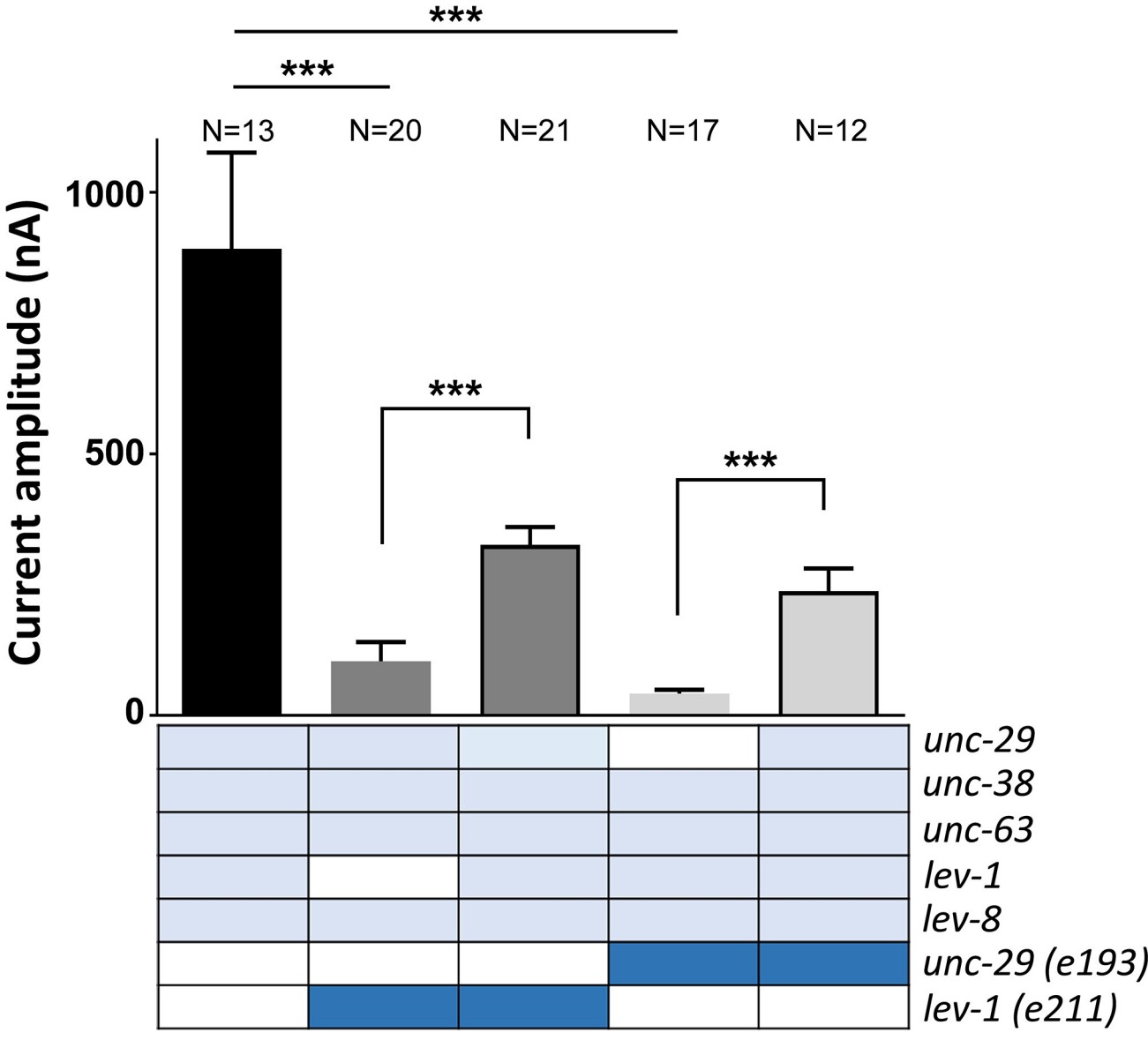

**Fig 9. The mutant genes in *lev-1 (e211)* and *unc-29 (e193)* encode for subunits that assemble L-type receptors with reduced function.** Current amplitude to 300 µM of acetylcholine was quantified in different populations of L-type receptors in *Xenopus oocytes* expressed from cRNA encoding either wild type, *lev-1 (e211)* or *unc-29 (e193)* mutations (dark blue). The substitution of the cRNA for wild type *lev-1* or *unc-29* subunits for their respective *lev-1 (e211)* or *unc-29 (e193)* mutations reduced the current amplitude of the L-type receptor. The co-expression of wild type and mutant genes in a 1:1 ratio causes a reduction of the amplitude to acetylcholine-evoked currents, indicating both subunits compete for inclusion into the mature ion channel. Data are shown as the mean ± SEM. Numbers above bars indicate the number of oocytes recorded for each condition. Statistical significance was calculated by two-tail t-test. ***p≤0.001.

10). The pharyngeal pumping was completely inhibited after 3 hours incubation with the drug. However, these worms did not show the within drug recovery of pumping in the presence of paraoxon (Fig 10). Both mutants were deficient in the paraoxon-induced recovery of function observed in the face of ongoing drug exposure as described for the N2 worms. This supports the notion that auxiliary organization impacting the structure or function of receptor signalling may underpin these observed changes in the efficacy of organophosphate intoxication.

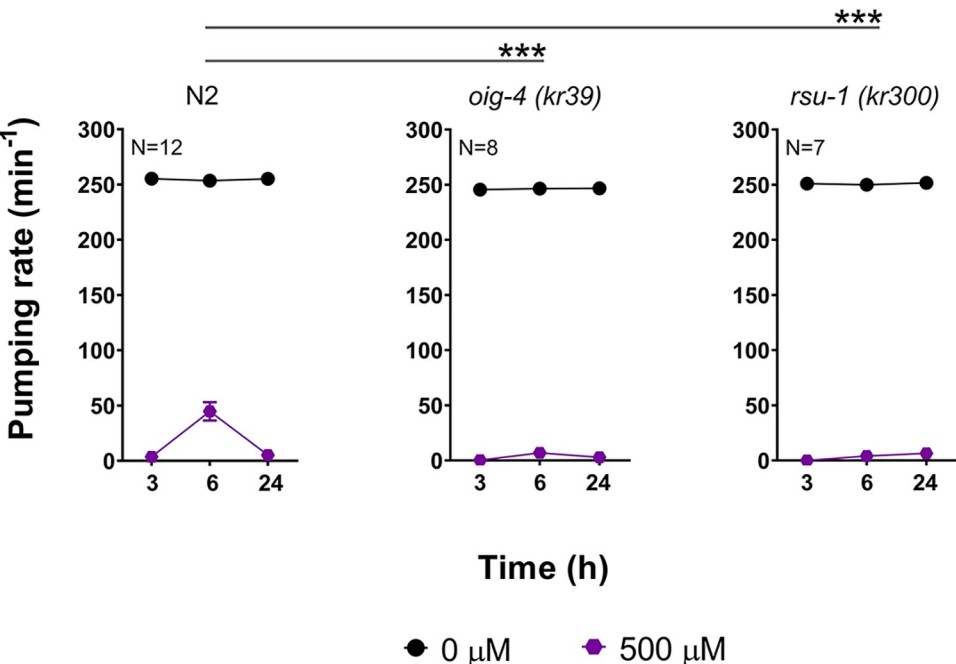

**Fig 10. Mutants that underpin the synaptic organization of the L-type receptors modifies the pharyngeal pumping observed in nematodes exposed to paraoxon-ethyl.** The pharyngeal function of wild type nematodes exposed to paraoxon-ethyl exhibited drug-induced plasticity. Data are shown as mean ± SEM of 12 worms in at least 6 independent experiments (N = 12). OIG-4 is a muscle-secreted protein involved in the location of the L-type receptor at the body wall NMJ [55]. *Oig-4* lacking nematodes are deficient in the in-drug recovery of pharyngeal pumping observed in the wild type worms. Data are shown as mean ± SEM of 8 worms in at least 4 independent experiments (N = 8). RSU-1 is a cytosolic muscle protein involved in maintaining the equilibrium between synaptic and extra-synaptic nicotinic receptors [54]. *rsu-1* lacking nematodes exposed to paraoxon-ethyl are deficient in the within drug recovery of the pharyngeal pumping. Data are shown as mean ± SEM of 7 worms in at least 4 independent experiments (N = 7). Statistical significance was calculated by two-way ANOVA test followed by Bonferroni corrections. ***p≤0.001.

## Discussion

Organophosphates are environmental biohazards that cause at least two million of poisoning cases and lead to an estimated 200,000 deaths annually [11, 12, 56]. The intoxication impedes of termination of the acetylcholine signalling because it prevents the acetylcholinesterase activity that breaks down neurotransmitter [1, 2]. The overstimulation of the cholinergic transmission in central and peripheral nervous systems triggers a wide range of clinical manifestations [1, 5]. However, the pharmacological treatment to mitigate the symptoms of the cholinergic syndrome is restricted to two main mechanisms, the inhibition of muscarinic receptors by atropine and the reactivation of organophosphate-bond acetylcholinesterase by oximes. Benzodiazepines are additionally used to treat seizures during the first stage of intoxication [6]. Since the efficiency of these medications is limited [6, 11, 57, 58], we propose the investigation of plasticity-promoting mechanisms in order to identify other targets to mitigate against the effects of anti-cholinesterase poisoning (Fig 11). The model organism *C. elegans* can be utilized for this purpose. This free-living nematode has neuromuscular function organized by a highly conserved cholinergic pathway that triggers easy quantifiable phenotypes [59–61]. In previous investigations, we highlighted the quantification of pharyngeal pumping movements as a suitable cholinergic-dependent phenotype to investigate acetylcholinesterase intoxication and recovery [36]. Here, we demonstrated that *C. elegans* are able to develop cholinergic plasticity

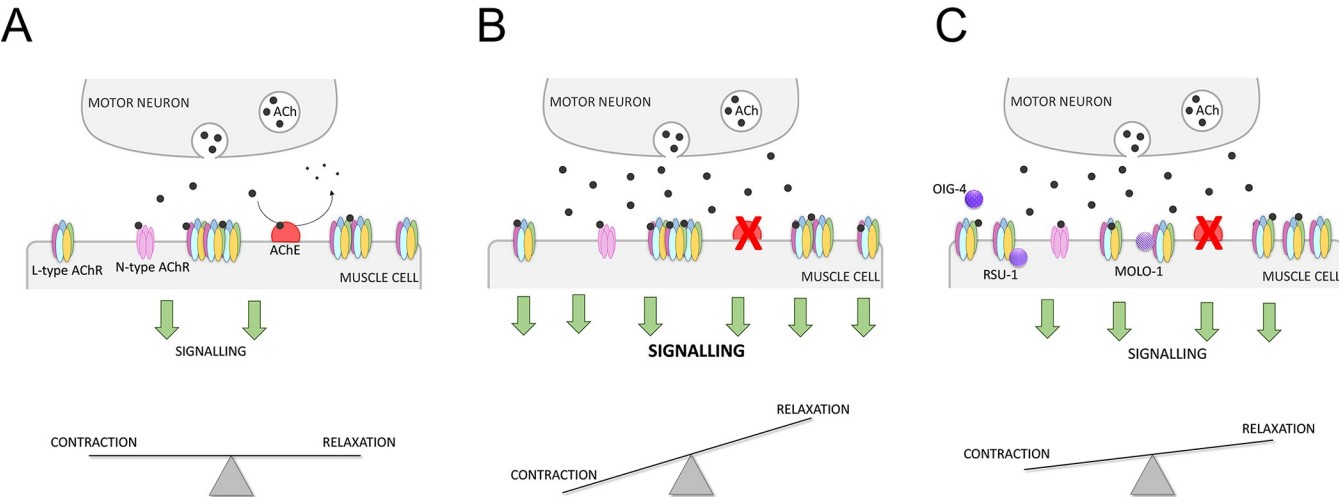

**Fig 11. Hypothesised mechanism underpinning paraoxon-induced plasticity in nematodes exposed to 500 µM paraoxon-ethyl.** A) The body wall motor neuron releases acetylcholine to the neuromuscular junction that activates L-type and N-type nicotinic receptors causing the muscle contraction. In the current study this determinant is measured by scoring pharyngeal function as body wall muscle contraction leads to an associated inhibition of pharyngeal function. At the level of the body wall muscle acetylcholinesterases catalyse the breakdown of acetylcholine. This controls cholinergic transmission from the motor neuron that allows balanced excitation at the body wall muscle. B) The inhibition of the acetylcholinesterase by paraoxon-ethyl causes an increase of acetylcholine at the neuromuscular junction. This leads to the overstimulation of the cholinergic receptors expressed in the muscle fibres and, therefore the hypercontraction of the body wall muscle and an associated inhibition of pharyngeal pumping. C) Molecular determinants that impact the location and sensitivity of L-type receptors at the body wall neuromuscular junction alter the consequence of organophosphate intoxication. Further investigations are required to identify if these determinants encompassing hypo-functional L-type receptors, MOLO-1, OIG-4, RSU-1 execute their impact thought structural or functional changes of the cholinergic synapse.

in two different contexts, preconditioning with low doses of the drug and during chronic exposure to relatively high concentrations.

The preconditioning with relatively low doses of paraoxon-ethyl increased sensitivity to the pharyngeal inhibition in nematodes post-exposed to higher concentrations of the same drug (Fig 2). Surprisingly, this effect seen with the organophosphate paraoxon-ethyl was completely opposite to the effect observed with the carbamate aldicarb (Fig 1). Organophosphates and carbamates have similar primary modes of action [2]. However, our data indicate that consequences of the intoxication beyond the inhibition of acetylcholinesterase are different for both chemicals.

In the context of chronic exposure to a high concentration of the organophosphate paraoxon-ethyl, nematodes developed mitigating plasticity in the presence of the drug. This was observed by a within drug treatment recovery of about 20% of the pharyngeal and body wall phenotypes after an initial complete initial inhibition by paraoxon-ethyl intoxication. The recovery of the pumping rate and body length was not sustained over the time and reverted to a more complete inhibition after 24 hours of continuous exposure (Fig 4). This indicates three phases to the intoxication response when nematodes are exposed to paraoxon-ethyl. An initial inhibition, followed by a recovery and a subsequent inhibition of both pharyngeal pumping and body length.

In order to identify molecular components of this potentially functional mitigating recovery, we systematically investigated the ability to express these three phases of paraoxon-ethyl intoxication in the pharyngeal pumping of mutants involved in the organization of acetylcholine synaptic signalling. We demonstrated that cholinergic components of the pharyngeal neuromuscular junction had little effect on the expression or modulation of the drug-induced recovery observed in the wild type nematodes (Fig 5). This builds on previous observations

indicating that the *per se* inhibition of the pumping rate in the presence of the anti-cholinesterase aldicarb is mediated by determinants executing their function outside the pharyngeal system and within the body wall muscle [37]. Indeed, the investigation with mutants in molecular components of the body wall neuromuscular junction highlighted the pivotal role of the L-type receptor signalling in modulating the paraoxon-ethyl mitigating plasticity of the pharyngeal pumping (Fig 11). This acetylcholine-gated cation channel is composed by the alpha subunits UNC-63, UNC-38 and LEV-8 and the non-alpha subunits UNC-29 and LEV-1 [47]. Auxiliary and ancillary proteins have been linked with the function of the receptor by controlling receptor trafficking, sensitivity, expression or clustering [24, 47, 54, 55, 62–65]. We observed that *oig-4 (kr39)* and *rsu-1 (kr300)* deficient nematodes lack the spontaneous recovery of the pharyngeal pumping in the presence of paraoxon-ethyl compared with wild type worms (Fig 10). OIG-4 is a muscle secreted protein that interacts with the complex formed by L-type receptor. RSU-1 is a protein expressed in the cytoplasm of the muscle cells and is required for the proper balance of the L-type receptor distribution between synaptic and extra synaptic regions [54]. The fact that these two proteins involved in the location of the muscle receptor might suggest that the position of the receptor during organophosphate intoxication could be altered to modulate the excess of cholinergic signal (Fig 11). This speculated alteration would be necessary for the expression of paraoxon-induced mitigating plasticity of the pharyngeal pumping. Further investigations are required to explicitly investigate if sub-cellular organization of receptors is important in the paradigm described here. However, previous observations have highlighted the altered distribution of nicotinic receptors at the mammalian neuromuscular junction of skeletal muscle in acetylcholinesterase knockout mice compensates for the chronic absence of this enzyme activity [66].

The two non-alpha subunits, UNC-29 and LEV-1, of the L-type receptor were previously pointed as pharmacological determinants of the aldicarb sensitivity in the pharyngeal pumping of nematodes exposed to the carbamate [37]. Here, we observed that mutant nematodes *unc-29 (e193)* and *lev-1 (e211)* displayed a paraoxon-induced response in the pharyngeal pumping characterized by only two phases: the initial inhibition and a sustained recovery in the face of on-going drug exposure. This recovery lacks the subsequent inhibition observed in wild type worms (Fig 6). Interestingly, this phenotype is not conserved in null mutant strains of *unc-29 (e1072)* or *lev-1 (x427)* but do phenocopy the response observed in *molo-1 (kr100)* deficient nematodes (Fig 8).

MOLO-1 is an auxiliary protein involved in the sensitivity of the L-type acetylcholine receptor [24]. Thus, the similarity in responses from receptors encoded by subunits derived from the *e193* mutation of *unc-29* and the *e211* mutation of *lev-1* indicates they have a change in receptor function that may well lie at the level of agonist efficacy. In addition, we demonstrated that LEV-1 and UNC-29 subunits harbouring the *e211* and *e193* mutations, respectively, might be able to assemble with the alpha subunits UNC-38, UNC-63 and LEV-8, and reconstitute functional L-type receptors when expressed in *Xenopus oocytes*. The resulting L-type receptor displayed reduced current amplitude to a high concentration of acetylcholine (Fig 9) suggesting a role of the mutated subunits as modulators of the L-type receptor function. The fact that the *e193* mutation in UNC-29 exhibited a stronger dominant negative effect compared to *e211* in LEV-1 might help explain why this receptor subunit is less able to rescue of the phenotype observed in transgenic lines expressing wild type UNC-29 into the body wall muscle of a *unc-29 (e193)* mutant strain (Fig 7B). However, this less marked rescue might be underpinned by additional expression of UNC-29 outside the body wall muscle [67]. Further investigations are required to research this possibility.

The *e211* mutation consists in a glycine to glutamic acid substitution at the fourth transmembrane domain of the LEV-1 subunit (S1 Fig). This domain of the acetylcholine receptor

subunits mediates the interaction of the receptor with the phospholipid bilayer and contributes to the kinetics of activation of the receptor. Several studies revealed that mutations in the fourth transmembrane domain of the nicotinic receptor subunits alter the channel opening and closing more than the trafficking or expression of the receptor [67–71]. In fact, the equivalent mutation within the transmembrane domain of a non-alpha subunit in *Torpedo californica* reduced the potency of the receptor to high concentrations of acetylcholine but has no effect on the comparative receptor function at low doses of agonist [68]. This could explain why nematodes harbouring this change exhibit wild type phenotype for locomotion and pumping but strong resistance to inhibition of these two behaviours in the presence of anti-cholinesterases which drive high concentrations of synaptic acetylcholine [37].

The *e193* mutation of *unc-29* consists of a proline to serine substitution at the loop connecting the second and third transmembrane domain (S2 Fig). This proline is highly conserved in all the subunits of the cys-loop receptor superfamily of the ligand-gated ion channels [72, 73]. The coupling of this proline with extra cellular domains in the mammalian muscle-type receptor is essential for the gating of the channel without affecting the trafficking or expression [72, 74–77].

Four strains were identified by exhibiting a sustained recovery of the pharyngeal function in the presence of paraoxon-ethyl, *ric-3 (hm9)*, *lev-1 (e211)*, *unc-29 (e193)* and *molo-1 (kr100)*. RIC-3 is an ancillary protein responsible of the maturation of different types of acetylcholine receptors (L-type, N-type and ACR-2R) [47, 78]. This points that the disruption of the L-type receptor function during organophosphate intoxication might be a viable route to mitigate the symptoms of the cholinergic syndrome (Fig 11). The use of nicotinic receptor antagonists has been previously proposed in the treatment of organophosphate intoxication [79–82]. However, this medication causes a significant hypotension due to the blockage of the cholinergic signal at the parasympathetic ganglia [80, 81]. A more specific antagonist of the nicotinic receptor at the skeletal muscle has been considered. In fact, there is strong evidence supporting the idea that some oximes have a beneficial effect in the recovery from nicotinic overstimulation symptoms due to the blockage of the nicotinic receptors [83–85]. However, the allosteric modulation of the nicotinic receptor sensitivity which is less well studied could represent an additional attractive option [79–82]. Negative allosteric modulators that reduce the nicotinic receptor activation more as the stimulation by acetylcholine increases would be particularly compelling. Interestingly, non-competitive antagonist drugs have been demonstrated to block the open channel in both *in vivo* and *in vitro* organophosphate poisoning models with concomitant beneficial effects in the recovery from intoxication [82, 86–88].

Although further investigations will be required to identify pharmacological treatments to modulate the cholinergic signalling during organophosphate intoxication, our research open new insights into the mechanisms that induce cholinergic plasticity in this context (Fig 11). The nematode *C. elegans* might provide an attractive *in vivo* model for the screening of nicotinic receptor antagonists or modulators with potential effects against organophosphate poisoning.

## Materials and methods

### *C. elegans* maintenance and strains

Nematodes were maintained according to standard procedures [89]. Briefly, nematodes strains were grown at 20°C on NGM plates seeded with *E. coli* OP50 as source of food. Mutant strains EN39 *oig-4 (kr39)* II, EN300 *rsu-1 (kr300)* III and EN100 *molo-1 (kr100)* III were kindly provided by Jean-Louis Bessereau Lab (Institut NeuroMyoGène, France). ZZ427 *lev-1 (x427)* IV was kindly provided by William Schafer Lab (MRC Laboratory of Molecular Biology, UK).

The transgenic lines VLP1: CB211 *lev-1 (e211)* IV; *Ex[Punc-122::gfp]*; VLP2: CB211 *lev-1 (e211)* IV; *Ex[Punc-122::gfp; Pmyo-3::lev-1]* were previously available in the laboratory stock [37]. The following strains were acquired from CGC: N2 wild type, DA465 *eat-2 (ad465)* II, VC670 *gar-3 (gk337)* V, PR1300 *ace-3 (dc2)* II, MF200 *ric-3 (hm9)* IV, CB211 *lev-1 (e211)* IV, CB193 *unc-29 (e193)* I, CB1072 *unc-29 (e1072)* I.

The following transgenic lines were generated in this work: VLP10: CB193 *unc-29 (e193)* I; Ex[*Punc-122::gfp*]; VLP11: CB193 *unc-29 (e193)* I; Ex[*Punc-122::gfp; Pmyo-3::unc-29*].

### Generation of *unc-29* rescue constructs

The genomic region corresponding to 3.8 kb of *unc-29 locus* was amplified using the forward and reverse primers 5'- CAGATCTCTTATGAGGACCAACCGAC -3' and 5'- CTCTCA AAGTCAAAAAAAGGCGAGGAG -3' (58°C annealing temperature), respectively. The PCR product was sub-cloned into pCR8/GW/TOPO following the manufacturer protocol and sub-sequently cloned into pWormgate plasmid containing 2.3 kb of *myo-3* promoter [37]

PCR amplifications were performed using Phusion High-Fidelity PCR Master Mix with HF Buffer (Thermo Fisher Scientific) following manufacturer instructions.

### Generation of transgenic lines

The marker plasmid that drive GFP expression in coelomocytes was a kind gift by the Antonio Miranda Lab (Instituto de Biomedicina de Sevilla, Spain [90]).

Control and rescue transgenic lines of *unc-29* were generated by microinjection of the corresponding plasmids into one day old adults of CB193 *unc-29 (e193)* mutant strain [91]. A concentration of 50 ng/μl of the marker plasmid *Punc-122::gfp* was injected to generate the transgenic strain VLP10. A mixture of 50 ng/μl of *Punc-122::gfp* plasmid and 50 ng/μl of *Pmyo-3::unc-29* plasmid was microinjected to generate the transgenic strain VLP11.

The genotype of CB193 strain was authenticated by PCR amplification of the *unc-29 locus* and subsequent sequencing of the PCR product was carried out before microinjection.

### Sequencing of mutant alleles

Mutations in the *lev-1* and *unc*-29 mutant strains were analysed by PCR amplification (Table 1) of the corresponding genomic fragment followed by Sanger sequencing. RT-PCR was additionally performed to describe mutation in CB1072 *unc-29 (e1072)* strain following previously published protocols [37]. Briefly, a single worm from either CB1072 or N2 wild type strain was lysed and subsequently used for cDNA synthesis using SuperScript[TM] III Reverse Transcriptase kit in a total volume of 20 μl following manufacturer protocol (Invitrogen[TM]). 5 μl of the resulting cDNA was added to a final volume of 20 μl PCR reaction with indicated oligo primers (Table 1).

### Generation of *lev-1* and *unc-29* mutant cRNAs

*C. elegans lev-1* and *unc-29* cDNAs were cloned into the pTB207 expression vector. This vector has previously reported as suitable for transcription *in vitro* [47]. The *e211* (G461E) and the *e193* (S258P) mutations were respectively inserted in the LEV-1 and UNC-29 subunits by PCR using the Q5 site-directed mutagenesis kit according to the manufacturer's recommendations (New England Biolabs). The forward and reverse primers used were 5'- GTTCTTTGAGGC AACAGTTGG -3' / 5'- CCGTACAACAAAAACCGATCCA -3' for G461E substitution in *lev-1* cDNA, and 5' ATTCTT<u>TCA</u>CCAACATCTTCTACA -3' / 5'- CTTTGATACAAGAA GCAAGAACAC -3' for S258P substitution in *unc-29* cDNA. Underlined sequences indicate

**Table 1.** Primer sequences and PCR conditions for mutation analysis of alleles in *lev-1* and *unc-29*.

| Allele | Sample | Primers | Annealing temperature (˚C) | Amplification product (pb) |
|---|---|---|---|---|
| *e193* | gDNA | 5'– GGTATTTGGAAGTTGGACTGTG –3'<br>5'– GCTCAGATGCCGATTTTGGG –3' | 56 | 752 |
| *e1072* | gDNA | 5'– CAGATCTCTTATGAGGACCAACCGAC –3'<br>5'– CTCTCAAAGTCAAAAAAAGGCGAGGAG –3' | 58 | 3,870 |
| *e1072* | cDNA | 5'– ATTCTCTCATTCAGCCAGTCC –3'<br>5'– GCTCAGATGCCGATTTTGGG –3' | 55 | 937 |
| *e211* | gDNA | 5'– TGAAATAGAAAACGTGGGGG –3'<br>5'– AAAAGTTGAAAATGAAAGAATAATGG –3' | 58 | 965 |
| *x427* | gDNA | 5' AGAGAGAATGATGTTAGGAGG 3'<br>5' AGTTGAAAATGAAAGAATAATGG 3' | 55 | 4,940 |

PCR amplifications were performed employing Phusion High-Fidelity PCR Master Mix with HF Buffer (Thermo Scientific™) following manufacturer's recommendations.

the mutated codons. The resulting mutant clones were sequence-checked prior linearization (Eurofins Genomics). Respective wild-type and mutant subunit cRNAs were synthesized *in vitro* with the mMessage mMachine T7 transcription kit (Invitrogen), titrated and checked for integrity. Mixes of cRNAs containing 50 ng/μL of each cRNA encoding the *C. elegans* levamisole-sensitive acetylcholine receptor (UNC-63, UNC-38, UNC-29, LEV-1 and LEV-8) subunits of interest and ancillary factors (RIC-3, UNC-50 and UNC-74) were prepared in RNase-free water [47].

## Oocyte electrophysiology

To investigate the functional expression of the mutated LEV-1 or UNC-29 subunits, *C. elegans* L-type nicotinic receptors, either with wild type or modified subunits, were reconstituted in *Xenopus laevis* oocytes and assayed under voltage-clamp as previously described [92]. Briefly, 36 nl of cRNA mix were microinjected in defolliculated *Xenopus* oocytes (Ecocyte Bioscience) using a Nanoject II microinjector

(Drummond). After 4 days incubation, BAPTA-AM-treated oocytes were voltage-clamped at a holding potential of -60 mV and electrophysiological recordings were carried out as described previously [92]. Whole cell acetylcholine current responses were collected and analysed using the pCLAMP 10.4 package (Molecular Devices).

## Drug stocks

Aldicarb (Cas n. 116-06-3) and paraoxon-ethyl (Cas n. 311-45-5) were acquired from Merck. Aldicarb was dissolved in 70% ethanol and paraoxon-ethyl was dissolved in 100% DMSO in a stock concentration of 250 mM and 1M, respectively. The drug stocks were kept at 4˚C and used within one month or discarded. Obidoxime was provided by Dstl Porton Down (UK) and dissolved in distilled autoclaved water directly before use. Acetylcholine was purchased from Merck and dissolved in recording buffer (100 mM NaCl, 2.5 mM KCl, 1 mM CaCl$_2$.2H$_2$O, 5 mM HEPES, pH 7.3)

## Assay plates preparation

Anti-cholinesterase and obidoxime plates were prepared as previously described [36, 37] Briefly, assay plates were made by adding a 1:1000 aliquot of the more concentrated drug stock to the molten but tempered NGM agar to obtain the indicated concentration of either aldicarb

(50 μM and 250 μM) or paraoxon-ethyl (20 μM to 1 mM). 3 ml of the NGM containing the drug or the vehicle control was poured in each well of 6-well plates. After the agar solidified, plates were supplemented with 50 μl of *E. coli* OP50 ($OD_{600}$ 1) to act as the food source. The bacterial lawn was dried on the assay plates by incubating for 1 hour in a laminar flow hood. Assay plates were finally maintained at 4˚C in dark overnight. Plates were used within one day of being prepared and left at room temperature for at least 30 min before starting the experiment. There was no observable change in the bacterial lawn of drugged and control plates, therefore no effect of the anti-cholinesterase on the *E. coli* growth was discernible at any of the concentrations tested [93].

The final concentration of vehicle in the drug-containing and control plates was 0.07% ethanol for aldicarb assay plates and 0.1% DMSO for paraoxon-ethyl assay plates. Neither vehicle concentrations affected the phenotypes tested.

## Behavioural assays

Behavioural experiments were performed at room temperature (20˚C).

Pharyngeal pump rate on food was quantified by visual observation under a Nikon SMZ800 binocular microscope, using a timer and a handheld counter. The pumping rate was defined by the number of grinder movements per minute per worm. The pump rate was quantified for a minimum of 3 times for 1 minute each and the mean was used as pumps per minute.

The body length was measured as previously described [36]. Briefly, images of the worms were acquired through a Hamamatsu Photonics camera and visualized for recording with IC capture software (The Imaging Source©). These images were binarized and skeletonized using ImageJ software. The length of the skeleton was used to determine the body length of the nematodes.

## Preconditioning experiments with acetylcholinesterase inhibitors

Synchronized L4 stage worms were incubated on the preconditioning plates containing either drug or vehicle control. The concentration of acetylcholinesterase inhibitor used for preconditioning was selected as one that decreased the pharyngeal pumping rate to half of the maximal response following 24 hours exposure. This was 50 μM for aldicarb and 20 μM for paraoxon-ethyl {Izquierdo, 2020 #520}. After the pre-conditioning, nematodes were transferred on non-drug containing plates to allow the recovery of the pharyngeal function for 2 hours or 3 hours for aldicarb or paraoxon-ethyl preconditioned worms, respectively. Obidoxime plates were used during the recovery step from paraoxon-ethyl intoxication to facilitate the recovery of acetylcholinesterase after the drug inhibition [36]. Finally, nematodes were picked onto plates containing five times the concentration of drug used in the preceding preconditioning step (250 μM aldicarb or 100 μM paraoxon-ethyl). The pumping rate was measured at the indicated point times (10 min, 1, 3, 6 and 24 hours) after transferring the worms to the final control or drug treated observation plate.

## Protracted intoxication with paraoxon-ethyl

Synchronized L4 plus 1 stage nematodes were picked to either paraoxon-ethyl or vehicle control plates. Pharyngeal pump rate and body length was quantified at specified times after transferring to the assay plates (10 min, 1, 3, 6 and 24 hours). Nematodes often leave the patch of food during the first hour after transferring on paraoxon-containing plates. They were picked back to the bacterial lawn at least 10 minutes before pump rate was measured.

## Statistical analysis

The data were collected from control paired blind trials in which the experimenter was unaware of the genotype tested in each trial.

Data were analysed using GraphPad Prism 8 and are displayed as mean ± standard error of the mean (SEM) of the number of worms used for each assay. The sample size N of each experiment is specified in the corresponding figure and indicate the number of worms assessed. Statistical significance was assessed using two-way ANOVA followed by post hoc analysis with Bonferroni corrections where applicable. This post hoc test was selected among others to avoid false positives. Statistical significance for the electrophysiology data was calculated by two-tail t-test.

## Supporting information

**S1 Fig. *lev-1* mutant alleles of strains CB211 and ZZ427.** Genomic organization of *lev-1 locus* indicating the position of the single point mutation in *e211* and the deletion in *x427* alleles. CB211 *lev-1 (e211)* mutant strain contained a G to A missense mutation identified in exon 7 of the genomic DNA. This leads to a glycine to glutamic acid exchange at the fourth transmembrane domain (M4). The strain ZZ427 *lev-1 (x427)* contains a deletion of 1,267 bp from intron 3 to intron 4 and a T insertion. This causes a LEV-1 protein lacking the first, second and third transmembrane domain. Black arrow represents 100 bp and the sense of transcription. Black triangles in the genomic DNA represents the position of the four transmembrane domains. Chromatograms corresponds to the 5' to 3' readout of the minus strand. The position indicated in each chromatogram corresponds to the position of the respective base from the ATG starting codon in the genomic DNA of N2 wild type.
(JPG)

**S2 Fig. *unc-29* mutant alleles of strains CB193 and CB1072.** A) Genomic organization of *unc-29 locus* indicating the position of the single point mutation in *e193* and *e1072* alleles. CB193 *unc-29 (e193)* mutant strain contained a C to T missense mutation identified in exon 9 of the genomic DNA. This causes the exchange of conserved proline to serine at a conserved residue within the extracellular loop that connects the second (M2) and third (M3) transmembrane domain of the UNC-29 subunit. The strain CB1072 *unc-29 (e1072)* contains a G to A single point mutation identified in the splicing acceptor of intron 8 in the genomic DNA. This caused the formation of a new splicing site utilizing the first G in exon 9 (E9). B) RNA organization of *unc-29* indicating the position of the single point mutation in *e193* and *e1072* alleles. The mutation of *e1072* allele causes frameshift leading to a premature stop codon at the end of the predicted intracellular loop that connects the first (M1) and second (M2) transmembrane domains. Black arrow represents 100 bp and the sense of transcription. Black triangles in the genomic and cDNA represents the position of the four transmembrane domains. The position indicated in each chromatogram corresponds to the position of the respective base from the ATG starting codon in the genomic (A) or cDNA (B) of N2 wild type.
(JPG)

**S1 Data.**
(PZFX)

## Acknowledgments

We thank Dr Jean-Louis Bessereau, Laure Granger, Dr Denise Walker and Dr William Schafer for sharing strains; Dr Antonio Miranda-Vizuete for sharing *Punc-122*::*gfp* marker plasmid.

Additional *C. elegans* strains were provided by the CGC, which is funded by NIH Office of Research Infrastructure Programs (P40 OD010440).

## Author Contributions

**Conceptualization:** Patricia G. Izquierdo, Cedric Neveu, A. Christopher Green, John E. H. Tattersall, Lindy Holden-Dye, Vincent O'Connor.

**Data curation:** Claude L. Charvet.

**Formal analysis:** Patricia G. Izquierdo, Claude L. Charvet, Cedric Neveu, Vincent O'Connor.

**Funding acquisition:** A. Christopher Green, Lindy Holden-Dye, Vincent O'Connor.

**Investigation:** Patricia G. Izquierdo, Claude L. Charvet, Vincent O'Connor.

**Methodology:** Patricia G. Izquierdo.

**Project administration:** A. Christopher Green, Lindy Holden-Dye, Vincent O'Connor.

**Resources:** Cedric Neveu, John E. H. Tattersall.

**Supervision:** John E. H. Tattersall, Lindy Holden-Dye, Vincent O'Connor.

**Visualization:** Patricia G. Izquierdo.

**Writing – original draft:** Patricia G. Izquierdo.

**Writing – review & editing:** Patricia G. Izquierdo, Claude L. Charvet, Cedric Neveu, A. Christopher Green, John E. H. Tattersall, Lindy Holden-Dye, Vincent O'Connor.

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
