## [Decision Letter · Decision Letter 0]

11 Oct 2022

PONE-D-22-22312Organophosphate intoxication in C. elegans reveals a new route to mitigate poisoning through the modulation of determinants responsible for nicotinic acetylcholine receptor functionPLOS ONE

Dear Dr. O'Connor,

Thank you for submitting your manuscript to PLOS ONE. After careful consideration, we feel that it has merit but does not fully meet PLOS ONE’s publication criteria as it currently stands. Therefore, we invite you to submit a revised version of the manuscript that addresses the points raised during the review process. Please submit your revised manuscript by Nov 25 2022 11:59PM. If you will need more time than this to complete your revisions, please reply to this message or contact the journal office at plosone@plos.org. Please include the following items when submitting your revised manuscript:A rebuttal letter that responds to each point raised by the academic editor and reviewer(s). You should upload this letter as a separate file labeled 'Response to Reviewers'.A marked-up copy of your manuscript that highlights changes made to the original version. You should upload this as a separate file labeled 'Revised Manuscript with Track Changes'.An unmarked version of your revised paper without tracked changes. You should upload this as a separate file labeled 'Manuscript'.

We look forward to receiving your revised manuscript.

Kind regards,

Hongkyun Kim

Academic Editor

PLOS ONE

Journal Requirements:

   "No authors have competing interest"

Additional Editor Comments:

Your manuscripts were reviewed by two experts in the field. Both reviewers raised several serious concerns that you should address. Particularly, the title and abstract do not align with the data, and the introduction is not really informative and distracting. I recommend rewriting these sections to strengthen the manuscript and better inform the reader. In addition, the number of trials and animals in experiments was confusingly described.

Reviewers' comments:

Reviewer's Responses to Questions

**Comments to the Author**

1. Is the manuscript technically sound, and do the data support the conclusions?

Reviewer #1: Partly

Reviewer #2: Yes

2. Has the statistical analysis been performed appropriately and rigorously? 

Reviewer #1: Yes

Reviewer #2: No

3. Have the authors made all data underlying the findings in their manuscript fully available?

Reviewer #1: Yes

Reviewer #2: Yes

4. Is the manuscript presented in an intelligible fashion and written in standard English?

Reviewer #1: Yes

Reviewer #2: Yes

5. Review Comments to the Author

Reviewer #1: Here the authors present an interesting study in which they examine the impact of preconditioning with a low concentration of two different acetylcholinesterase inhibitors (aldicarb, paraoxon-ethyl) on subsequent response to a higher concentration of the drugs on an end point plate. They discovered that aldicarb preconditioning resulted in reduced sensitivity to subsequent aldicarb exposure (at 6 and 24 hrs), while paraoxon-ethyl preconditioning resulted in increased sensitivity to subsequent paraoxon-ethyl exposure (at 3 hrs). This was striking, because these are both cholinesterase inhibitors. The authors then went on to examine adaptation to high concentrations of the organophosphate paraoxon-ethyl and discovered that after an initial abolishment of pharyngeal pumping, the animals partially recovered after 6 hours of exposure, though pumping entirely ceased again by or before 24 hours of exposure. Interestingly, partial loss of function mutations in subunits of the nAChR that reduced current amplitude, resulted in sustained recovery, such that the mutants exhibited pharyngeal pumping even after 24 hours of paraoxon-ethyl exposure. Site of action may be different for these subunits as body-wall muscle expression of lev-1 in the lev-1 mutant resulted in a phenotype indistinguishable from wild type. However, expression of unc-29 in the body wall muscles of the unc-29 mutant was not sufficient to inhibit the sustained recovery of pumping on the organophosphate plates. While the data are interesting, there are a few places where additional experiments could strengthen the conclusions. Further, significant changes should be made to the text to improve clarity and more accurately reflect the scope of the data.

Major Concerns

1) The paper is written in way such that the focus is on identifying and understanding molecular mechanisms that can be targeted to mitigate the toxic effects of organophosphates (acetylcholinesterase inhibitors more generally) and the authors look at nAChRs. However, nAChR antagonists have already been proposed as a treatment in response to organophosphate exposure, which the authors do an excellent job of discussing (lines 496-511). While the authors provide a future direction to specifically identify pharmacological treatments (lines 513-517), the current paper does not explore specific nAChR antagonists. While the data presented are interesting, the paper should be framed somewhat differently such that the Title, Abstract, and text more specifically reflect the data that are presented in this paper.

2) Why do the lev-11(e211) data look so different in Figure 7 vs. Figure 8A vs. Figure 9? These animals were all exposed to the same concentration of paraoxon-ethyl. In addition, why do the lev-11(e211) data and lev-11(e211) control data (GFP expression in this mutant background) look so different in Figure 8A? Could this be due to small sample size? Plate variability? The inconsistency of these data is concerning.

3) OIG-4 and RSU-1 are minimally examined in this paper (Figure 11) and then discussed in relationship to data not shown for LEV-9 and LEV-10 (lines 430-450). I feel that these data go beyond the scope of the current work – Figure 11 is not needed for this to be a good paper. Figure 11 could be removed and then these data could be expanded on in a subsequent paper in which some of the authors' hypotheses about receptor levels and localization could be tested. Related to this, the authors don’t present data in their paper to support Figure 12C as they do not look at receptor localization in these mutants +/- paraoxon-ethyl. Rather than show a hypothesized model for which data doesn’t yet exist, this final figure panel would be stronger if it showed a model built from the data in the paper. If the authors choose to leave the Figure as it is, question marks should be present on the Figure itself.

4) Throughout the legends, it is unclear what n=. For example, when the authors state in Figure 1 that “Data are shown as mean +/- SEM of 6 worms in at least 3 independent experiments,” does this mean that total n=18 worms, or does this mean that 2 worms were assayed on 3 different days for a total n=6 worms. This needs to be clarified in all figures. In addition, if it is the latter, an n=6 is not an acceptable sample size. *** This is why I responded "partly" to the first question above, as I was unable to assess what the sample sizes were.

Minor concerns

1) When I think of plasticity, I think of the ability of a cell (in this case muscle) to change its structure or function. And I suppose that it what is happening here. However, the word “plasticity” is used so often and to refer to multiple different phenotypic changes that it becomes almost non-specific (for example, the title in line 178 is unclear). Use of more precise language in the Results as well as using wording such as “increased drug sensitivity” would greatly improve clarity of the paper.

2) Related to Figure 4, what is the impact of the reverse treatment - paraoxon-ethyl pre-treatment followed by aldicarb on the end point plate? If these data look similar to Figure 3A, this may suggest that paraoxon-ethyl has a second target in addition to acetylcholinesterase that sensitizes the animals to treatment with an acetylcholinesterase inhibitor on the end point plate.

3) In Figure 5, the authors show that paraoxon-ethyl induces body length shrinkage followed by partial recovery at 6 hrs, and a return to short length at 24 hours. This phenotype is very straightforward to quantitate and easier to measure than pharyngeal pumping. What are the unc-29 and lev-11 mutant phenotypes in the length measurement assay? Do you see a similar impact as observed for the pharyngeal pumping phenotype?

4) Expression of unc-29 in the body wall muscles does not impact the sustained recovery observed in the mutant. Does neuronal expression have an effect on this phenotype? This possibility could at least be discussed.

5) Line 319-321 (Figure 9): Looking at the data, lev-1(x437) looks like it may actually have recovery, and it may be sustained. What was the n= here? If only 8 worms, having a robust sample size could change your conclusion.

6) Line 354-358 (Figure 10): It’s important to indicate what you’re comparing to here. Co-expressing both WT and mutant subunits together results in greater current amplitude than having only the mutant subunit. This figure should include statistics, ie, compare bar 1 vs. bar 2 and bar 2 vs. bar 4

7) Line 362: You do not show any evidence that the mutant subunits change the sensitivity of the channel to the neurotransmitter. It’s possible that these subunits change the stability of the channel, abundance at the membrane, or the channel conductance, etc.

8) Lines 494-495: The authors do not show that LEV-1 (e211 mutant), UNC-29 (e193 mutant), and MOLO-1 affect channel gating – the impact was on current amplitude and this could have been for a number of reasons (and they do not investigate the impact of MOLO-1 on current amplitude)

9) In the Drug stocks and assay plates preparation sections of the methods, catalog numbers and stock concentrations should be indicated.

10) Statistical Analysis section in the methods should be updated to clearly reflect what n= and SEM means throughout the paper.

11) Additional changes to Figures:

a. It is unclear what SEM means. Is it the SEM of all worms assayed?

b. In Figure 1, the legend states **p≤0.01; ***p≤0.001, but there are no asterisks on the figure.

c. Figure 1 and Figure 2 could be merged

d. In Figures 2 and 3, it would be helpful to put the drug names on the figure as presented in Figure 4 (everywhere you have 20 uM or 100 uM, indicate what the drug is)

e. Italicize all gene names in Figure 8

Suggested Text Changes

Significant proofreading is needed; here are some (but not all) places where corrections should be made

Line 22: Do you mean insults or toxins (instead of disorders)?

Line 28: “Intrinsic determinants of receptor’s location and sensitivity modulate the extent of plasticity in the context of persistent cholinergic stimulation” don’t have enough data to state this

Line 31: Remove word novel as this has been tried before

Line 63: This sentence is unclear

Line 73: Missing the word “with”

Line 90: Egg laying is not required for reproduction

Line 99: Mediated is spelled wrong

Line 101: Did you mean to use the word exhibit instead of express?

Line 105: Did you mean to use the word high instead of large?

Lines 110-113: Remove word “new” and rewrite these sentences as nAChR antagonists have already been proposed in response to organophosphate exposure

Line 122: Include the words “in response” before “to the exposure….”

Line 127: Should read “either on” instead of “either in”

Line 158: For clarity, reword “aggravated inhibition of pumping” – possibly to “increased drug sensitivity”

Line 171: For clarity, reword “aggravated behavioral plasticity” – possibly to “enhanced sensitivity”

Line 192: Connection between first part of paper and second part of the paper should be clearer

Line 224: Sentence should be reworded

Line 231: Suggested wording change from “imply the expansion of treatments” to “suggest additional targets”

Line 281: italicize lev-1 and ric-3

Line 291: Wording change needed

Line 298: Need to add to this sentence; something like “transgenic expression of lev-11 and unc-29 in the body wall muscles of the lev-11(e211) and unc-29(e193) mutants, respectively.”

Line 333: Missing the word receptor

Line 342: Missing the word nAChR

Line 391: Instead of “develop alternative pathways” perhaps change to “identify other targets”

Line 396: Add the word “to”

Line 520: Add the word “to”

Line 521: Should be “NGM” instead of “NMG”

Line 528: Should be “CB1072” instead of “CB1071”

Line 955 – I think you mean “transferred” not “intoxicated”

Reviewer #2: Overall this study is carefully undertaken with appropriate controls and statistical data analysis although there are significant weaknesses that should be addressed

Major:

1. Oocyte electrophysiology (Fig. 10) lacks statistical analysis, and '*'s indicating significance are missing from Figure 1.

2. Fig. 8 is not well described. The 'control line' only makes sense when you dig into the Methods. The rationale for rescuing in body wall muscles is not stated. The reader must go back to the previous JBC paper to understand why this experiment was performed. Overall this figure needs to be much clearer and better explained.

3. cholinergic control of pharyngeal pumping should be described in the Introduction along with a discussion of Izquierdo, Callahorro et al JBC 2022. In fact, this paper is not listed in the bibliography. A summary of the major findings of this paper, in the Introduction, would help address point 2 above

4. there is no discussion of potential signaling mechanisms between the body wall muscles and the pharynx and how the specifics of ACh signaling might interact with that signaling pathway.

Minor - many writing and format flaws were noted which detract from the overall quality and readability of the manuscript. Below is a partial list. Thorough review is needed.

Text references to figures should indicate which part of the figure is being referenced (i.e. Fig. 7a, Fig. 7b)

Line 28 – receptor’s should be receptor

Line 99- ‘meadiated’ should be ‘mediated’

Line 105 – ‘incubation to’ should be ‘incubation with’ or ‘exposure to’

Fig. 1 stars indicating significant differences are not shown anywhere on figure

Line 143. Should make it clear in the previous sections that aldicarb is a carbamate acetylcholinesterase

Line 151 – should be ‘similar to the aldicarb experiment’

Line 154 – no support is provided for the statement that worms recover pumping 3 hr after removal from paraoxon-ethyl

178 – grammatical flaw in this heading, unclear meaning

254 – ‘spontaneous recovery even present’ meaning is unclear

255 ‘it’ should be ‘these results’

258-259 heading is confusing and needs to be reworded. Notably, no molecular imaging is shown, so the location of any protein cannot be stated on the basis of the data

288 – ‘function than’ should be ‘function as’

461-462 meaning is unclear: “shifting….influencing potency and/or efficacy”

955 – incubated on better than intoxicated onto

963 – “OP” abbreviation undefined

964 – incubated on better than incubated onto. Other instances of this flaw also noted (line 1007)

1079 – spontaneous recovery, not ‘spontaneously recovery’

Fig 7 – which comparison does the uppermost of the 3 bars across the top indicate? The end of the line does not neatly align with either the 6 or the 24 hr time point, but lies in between

Line 316 ‘pb’ should be ‘bp’

6. PLOS authors have the option to publish the peer review history of their article (what does this mean?). If published, this will include your full peer review and any attached files.

Reviewer #1: No

Reviewer #2: No

---

## [Author Response · Author response to Decision Letter 0]

9 Jan 2023

Dear Hongkyun Kim,

Thank you for yours and the expert referees’ review of our submission “Organophosphate intoxication in C. elegans reveals a new route to mitigate poisoning through the modulation of determinants responsible for nicotinic acetylcholine receptor function” (Ref PONE-D-22-22312). We found the comments expert and thorough. Given this, the changes we have made in accordance with the editorial and referee comments make for a better manuscript.

Overview response to referees general concerns.

Within the overview one referee commented in response to the question “Is the manuscript technically sound, and do the data support the conclusions?”: PARTLY.

We have responded to all comments of the referees in particularly issues around methodology and experimental numbers as specific points are raised. Having done this, we envisage we have made changes to re-assure a yes response to this issue from the referee.

In response to the question “Has the statistical analysis been performed appropriately and rigorously?” one referee responded with: NO.

We have clarified experimental numbers in response to specific referee comments. Further where the referees specifically requested statistical analysis we have performed these and modified the resubmission accordingly. Given this we envisage that this will have placated this referee’s concern.

Specific Editorial Requests.

The editor noted in their overview “Particularly, the title and abstract do not align with the data, and the introduction is not really informative and distracting. I recommend rewriting these sections to strengthen the manuscript and better inform the reader.”

With respect to this, we have used the prompts and specific queries to refine the text at the level of the title, abstract, introduction and the discussion. Where these aggregated correction are incorporated into the text we envisage they placate this general request for editing. We highlight where these changes have been made in the resubmitted text. 

The editor noted “In addition, the number of trials and animals in experiments was confusingly described.” This reinforces some specific points made by reviewer 1 that we have explicitly addressed and assume placate the more general editorial comment.

Specific responses to each referees in terms of “Comments to the Author” are mapped out in the listing below. We have addressed these by considering individual referees highlighting their comment in italics and our response below each reviewer query is shown in bold. To help with cross referencing we have structured comments around a numerical listing. We have resubmitted an unmarked manuscript and an additional marked up file in which yellow highlight is used to direct reviewers to points of major changes in text.

Reviewer #1 Comment 1

“The paper is written in way such that the focus is on identifying and understanding molecular mechanisms that can be targeted to mitigate the toxic effects of organophosphates (acetylcholinesterase inhibitors more generally) and the authors look at nAChRs. However, nAChR antagonists have already been proposed as a treatment in response to organophosphate exposure, which the authors do an excellent job of discussing (lines 496-511). While the authors provide a future direction to specifically identify pharmacological treatments (lines 513-517), the current paper does not explore specific nAChR antagonists. While the data presented are interesting, the paper should be framed somewhat differently such that the Title, Abstract, and text more specifically reflect the data that are presented in this paper”.

We agree with the referee and the sentiment that led to the editorial guidance. For this reason we have made important changes to the title, abstract and introduction. We have largely removed grandiose around broad descriptions of plasticity by removing text. We have focussed the functional readout from the results around the mitigation of functional inhibition seen even in the face of persistent drug exposure. We have stated in the introduction that pharyngeal pumping is a proxy for body wall muscle acetylcholine receptor function. This is done by describing and citing our own previous publication. This is highlighted in the introductory text to better guide the reader to the rationale of later experiments.

The resubmission includes: 

1. A new title “Modelling organophosphate intoxication in C. elegans highlights nicotinic acetylcholine receptor determinants that mitigate poisoning.” 

2. A new abstract.

3. A rearranged and focussed introduction.

4. Modifications to the discussion.

Reviewer 1 Comment 2 

“Why do the lev-11(e211) data look so different in Figure 7 vs. Figure 8A vs. Figure 9? These animals were all exposed to the same concentration of paraoxon-ethyl. In addition, why do the lev-11(e211) data and lev-11(e211) control data (GFP expression in this mutant background) look so different in Figure 8A? Could this be due to small sample size? Plate variability? The inconsistency of these data is concerning.”

Thank you for the comment. The referee is correct that there is a variation in the pattern of pharyngeal inhibition between graphs in lev-1 (e211) mutant, however the data is displayed to highlight the nature of the maintained function in the presence of the drug, and this is a consistent observation in all our experiments.

We control variation by using a defined experimental procedure which comprises of defined use of the age of the plate and the drug used, time in which the drug is on the plate before conducting the experiment and carefully staging worm development. Additionally, experiments were performed blind and always paired with the control. We consider that the key biological differences we have reported against the experiments are sound and sufficiently powered to allow the statements made.

The referee is correct with respect to the control line in lev-1 mutant background. The GFP expression in coelomocytes in this background appeared to modulate efficacy of intoxication at 24 hours-time point. Additional experiments beyond the scope of this research indicated a potential function of coelomocytes in drug toxicity. However, the functional rescue by the body wall muscle expression remains clear despite this confound. We have added a note in the corresponding result section to highlight the observation raised by the reviewer.

Reviewer 1 Comment 3 

“OIG-4 and RSU-1 are minimally examined in this paper (Figure 11) and then discussed in relationship to data not shown for LEV-9 and LEV-10 (lines 430-450). I feel that these data go beyond the scope of the current work – Figure 11 is not needed for this to be a good paper. Figure 11 could be removed and then these data could be expanded on in a subsequent paper in which some of the authors' hypotheses about receptor levels and localization could be tested. Related to this, the authors don’t present data in their paper to support Figure 12C as they do not look at receptor localization in these mutants +/- paraoxon-ethyl. Rather than show a hypothesized model for which data doesn’t yet exist, this final figure panel would be stronger if it showed a model built from the data in the paper. If the authors choose to leave the Figure as it is, question marks should be present on the Figure itself.”

We recognize the referee’s concern, and we understand that detailed textual speculation around the result description is not helpful for the manuscript’s key point. However, we prefer to show the data for OIG-4 and RSU-1 as it highlights that disruption and modulation of body wall muscle determinants can drive a mitigating or worsening functional outcome. We have shortened the discussion around these observations, removed the reference to unpublished data and modified the legend in Figure 12 to highlight that this aspect of the work requires further investigation. Essentially using the reviewer’s suggestion to highlight the speculation in this part of the discussion.

Reviewer 1 Comment 4

“Throughout the legends, it is unclear what n=. For example, when the authors state in Figure 1 that “Data are shown as mean +/- SEM of 6 worms in at least 3 independent experiments,” does this mean that total n=18 worms, or does this mean that 2 worms were assayed on 3 different days for a total n=6 worms. This needs to be clarified in all figures. In addition, if it is the latter, an n=6 is not an acceptable sample size. *** This is why I responded "partly" to the first question above, as I was unable to assess what the sample sizes were.”

Thank you for the comment. The N number was indicated in each graph, but we have made this clear in each of the figure legends and in the method section. We performed paired experiments which based on the required interleaving between control and treatment comparisons means we observe to 2 worms from each genotype at the different times recorded on a single plate. On occasion worms are censored individually observed worms disappear resulting in odd numbered N numbers.

Although we understand sample size is small for some of the experiments, Bonferroni corrections for statistical analysis was used to avoid false positive results.

Minor concerns Reviewer 1

Reviewer 1 Comment 5

“When I think of plasticity, I think of the ability of a cell (in this case muscle) to change its structure or function. And I suppose that it what is happening here. However, the word “plasticity” is used so often and to refer to multiple different phenotypic changes that it becomes almost non-specific (for example, the title in line 178 is unclear). Use of more precise language in the Results as well as using wording such as “increased drug sensitivity” would greatly improve clarity of the paper.”

Thank you for highlighting this. We agree with the referee’s sensibility to this point. We have constrained the use of plasticity and the associated grandiose. As the referee suggests, we have described the phenotypic observation in the literal “mitigation of organophosphate induced inhibition of pumping in the presence of drug”. This explicit better represents what we quantify and see as changing when using the pharynx as a bioassay of drug intoxication.

Reviewer 1 Comment 6 

“Related to Figure 4, what is the impact of the reverse treatment - paraoxon-ethyl pre-treatment followed by aldicarb on the end point plate? If these data look similar to Figure 3A, this may suggest that paraoxon-ethyl has a second target in addition to acetylcholinesterase that sensitizes the animals to treatment with an acetylcholinesterase inhibitor on the end point plate.”

The referee’s query is interesting; however this is beyond the scope of this investigation. The key point of this research was trying to find ways to improve the phenotypic response in the context of organophosphate intoxication either by preconditioning or chronic exposure. Since the preconditioning of paraoxon-ethyl aggravated the phenotypic response to the subsequent post-exposure, we did not consider the preconditioning of paraoxon-ethyl for subsequent experiments.

Reviewer 1 Comment 7

“In Figure 5, the authors show that paraoxon-ethyl induces body length shrinkage followed by partial recovery at 6 hrs, and a return to short length at 24 hours. This phenotype is very straightforward to quantitate and easier to measure than pharyngeal pumping. What are the unc-29 and lev-11 mutant phenotypes in the length measurement assay? Do you see a similar impact as observed for the pharyngeal pumping phenotype?”

The referee is correct. There is a good association between pharyngeal inhibition and body size (proxy for muscle contraction). The body length of the mutants analysed that were resistant to organophosphate and aldicarb intoxication was measured and have a reduced shrinkage in the presence of paraoxon-ethyl. The mechanism unpinning the cross talk between these two tissues remain unknown but pump inhibition in response to pharmacological activation of body wall muscle contraction is a good measure of function. This is discussed in our previous published article which we have made clear in the re-submission.

Reviewer 1 Comment 8

“Expression of unc-29 in the body wall muscles does not impact the sustained recovery observed in the mutant. Does neuronal expression have an effect on this phenotype? This possibility could at least be discussed.”

The referee is correct in the observation. The function of UNC-29 in other cells might additionally affect the phenotype we are presenting here. We have not performed the rescue experiment of unc-29 mutant expressing the wild type version of unc-29 under the control of its own promoter. However, the expression of the mutated subunit in oocytes supports that this mutated subunit can form functional channels with a strong dominant negative effect supporting the hypothesis of the function in the body wall muscle.

Additional text has been added to the discussion to include the hypothesis that expression beyond body wall muscle might contribute to outcomes observed in unc-29 mutants.

Reviewer 1 Comment 9

“Line 319-321 (Figure 9): Looking at the data, lev-1(x437) looks like it may actually have recovery, and it may be sustained. What was the n= here? If only 8 worms, having a robust sample size could change your conclusion.”

Thank you for the observation. There is the possibility that increasing the N number in lev-1 (x427) experiment we could conclude a significant sustained recovery of the pharyngeal function in the presence of paraoxon-ethyl (Figure 8). However, the most interesting comparison in this figure is the response to paraoxon-ethyl between the two lev-1 mutant backgrounds and the current paired analysis does not suggest any difference by increasing the N number in the experiment.

We have removed the text from the results section and the corresponding figure legend containing explicit reference of the lack of sustained recovery in this strain.

Reviewer 1 Comment 10

“Line 354-358 (Figure 10): It’s important to indicate what you’re comparing to here. Co-expressing both WT and mutant subunits together results in greater current amplitude than having only the mutant subunit. This figure should include statistics, ie, compare bar 1 vs. bar 2 and bar 2 vs. bar 4

Thank you for highlighting this. The corresponding part of the result section has been revised to better describe the relevant comparisons. In addition, figure and figure legend have been modified to include statistical comparisons where relevant.

Reviewer 1 Comment 11

“Line 362: You do not show any evidence that the mutant subunits change the sensitivity of the channel to the neurotransmitter. It’s possible that these subunits change the stability of the channel, abundance at the membrane, or the channel conductance, etc.”

The referee makes an important critique. Although the data is clear and mutated subunits play a role in the function of the resulting L-type receptor, this role might be ascribed to different explanations. For this reason, we have reworded the corresponding results section removing explicit observations about the sensitivity of the receptor to the neurotransmitter. 

Reviewer 1 Comment 12

8) Lines 494-495: The authors do not show that LEV-1 (e211 mutant), UNC-29 (e193 mutant), and MOLO-1 affect channel gating – the impact was on current amplitude and this could have been for a number of reasons (and they do not investigate the impact of MOLO-1 on current amplitude)

This is a fair point and we have reworded the mentioned sentence to be more precise.

Reviewer 1 Comment 13

In the Drug stocks and assay plates preparation sections of the methods, catalog numbers and stock concentrations should be indicated.

These details have been added in the methods section.

Reviewer 1 Comment 14

Statistical Analysis section in the methods should be updated to clearly reflect what n= and SEM means throughout the paper.

The experimental design is mapped out in which the number of observed animals in each independent paired experiment is defined. The statistical analysis section in the methods has been updated to include these details.

Reviewer 1 Comment 15

Additional changes to Figures:

a. “It is unclear what SEM means. Is it the SEM of all worms assayed?”

Yes. This point has been clarified in the statistical analysis section.

“b. In Figure 1, the legend states **p≤0.01; ***p≤0.001, but there are no asterisks on the figure.”

The missing asterisks have now been added to the corresponding graph which is now merged with figure 2 to create a new figure 1.

“c. Figure 1 and Figure 2 could be merged”

The referee makes a sensible request and we have combined the panels from these two figures.

“d. In Figures 2 and 3, it would be helpful to put the drug names on the figure as presented in Figure 4 (everywhere you have 20 uM or 100 uM, indicate what the drug is)”

We have made these changes to the figures as requested.

e. Italicize all gene names in Figure 8

Thank you for noticing this. The gene names have been italicized.

Reviewer 1 Suggested Text Changes

“Significant proofreading is needed; here are some (but not all) places where corrections should be made”

We have re-written and proofed the submission before replying.

Line 22: Do you mean insults or toxins (instead of disorders)?

The statement containing this point is removed to help provide a better focus for the abstract.

Line 28: “Intrinsic determinants of receptor’s location and sensitivity modulate the extent of plasticity in the context of persistent cholinergic stimulation” don’t have enough data to state this

In view of the referees comment on their preferred view we have encapsulated this using the description “receptor organization”.

Line 31: Remove word novel as this has been tried before

We have replaced the word novel with underexploited.

“Line 63: This sentence is unclear”

The sentence is part of a paragraph that detracts from the focus and based on referee and editorial advice has been removed.

Line 73: Missing the word “with”

Thank you. This has been added.

“Line 90: Egg laying is not required for reproduction”

Thank you. We have removed this sentence from the manuscript.

“Line 99: Mediated is spelled wrong”

Thank you. We have corrected this.

Line 101: Did you mean to use the word exhibit instead of express?

Thank you for the suggestion. We have used exhibit.

“Line 105: Did you mean to use the word high instead of large?”

Thank you. High is pharmacologically acceptable so we have added this change. 

Lines 110-113: Remove word “new” and rewrite these sentences as nAChR antagonists have already been proposed in response to organophosphate exposure

Thank you. We have removed this overstatement and suggest that this is an underexploited approach.

Line 122: Include the words “in response” before “to the exposure….”

We have made the suggested change.

Line 127: Should read “either on” instead of “either in”

Thank you. We have made the suggested change.

Line 158: For clarity, reword “aggravated inhibition of pumping” – possibly to “increased drug sensitivity”

We suggest a change in which “enhanced” replaces “aggravated”.

Line 171: For clarity, reword “aggravated behavioral plasticity” – possibly to “enhanced sensitivity”

We suggest “enhanced behavioural change”.

Line 192: Connection between first part of paper and second part of the paper should be clearer

Thank you. We agree and have used the simple link “In addition to investigate preconditioning, we studied if adaptation in C. elegans can follow upon the chronic exposure to exaggerated concentration of the stimulus” at this point in the text.

Line 224: Sentence should be reworded

Sentence has been reworded and now it reads “Furthermore, the similar recovery of cholinergic function for pumping rate and body length when nematodes are continuously exposed to the drug suggests a common underpinning mechanism.”

Line 231: Suggested wording change from “imply the expansion of treatments” to “suggest additional targets”

Thank you. We have made this suggested change.

Line 281: italicize lev-1 and ric-3

Thank you. This change has been made.

Line 291: Wording change needed

Thank you. We have removed the disrupting word “happened”.

Line 298: Need to add to this sentence; something like “transgenic expression of lev-11 and unc-29 in the body wall muscles of the lev-11(e211) and unc-29(e193) mutants, respectively.”

Thank you. We have made the suggested change that helps convey better clarity. 

Line 333: Missing the word receptor

Thank you. The word “receptor” has been added to the sentence.

Line 342: Missing the word nAChR

We have used “acetylcholine receptor” instead.

Line 391: Instead of “develop alternative pathways” perhaps change to “identify other targets”

Thank you. We have added the change to add precision.

Line 396: Add the word “to”

This sentence has been removed to reduce distracting grandiose.

Line 520: Add the word “to”

Thank you. The word “to” has been added to the sentence.

Line 521: Should be “NGM” instead of “NMG”

Thank you for highlighting this. This has been corrected.

Line 528: Should be “CB1072” instead of “CB1071”

Thank you. This mistake has been corrected.

Line 955 – I think you mean “transferred” not “intoxicated”

We have changed “intoxicated” by the word “incubated”

Reviewer 2

Major:

Reviewer 2 Comment 1.

“Oocyte electrophysiology (Fig. 10) lacks statistical analysis, and '*'s indicating significance are missing from Figure 1.”

Thank you for raising these points. 

We have now performed the statistical analysis on data from the oocyte electrophysiology experiment and added the corresponding text to the figure legend and methods. The figure has been modified to include the statistical analysis.

We have also added the missing asterisk to Figure 1. This figure has now been combined with figure 2 on request of the other reviewer.

Reviewer 2. Comment 2

“Fig. 8 is not well described. The 'control line' only makes sense when you dig into the Methods. The rationale for rescuing in body wall muscles is not stated. The reader must go back to the previous JBC paper to understand why this experiment was performed. Overall this figure needs to be much clearer and better explained.”

Apologies to the referee. We have now introduced the principle that body wall muscle receptor is the major determinant of the drug-induced inhibition of pharyngeal muscle within the introduction. This important flag makes it easier for the reader to distil the rationale for the body wall muscle expression used in Figure 8 (now Figure 7). The figure legend has also been updated to include the genotype of the transgenic lines presented in the graphs.

Reviewer 2 Comment 3

“cholinergic control of pharyngeal pumping should be described in the Introduction along with a discussion of Izquierdo, Callahorro et al JBC 2022. In fact, this paper is not listed in the bibliography. A summary of the major findings of this paper, in the Introduction, would help address point 2 above”

Yes. As addressed above (Reviewer 2 Comment 2) we recognize this and have modified the text at the points indicated by the reviewer.

Reviewer 2 Comment 4

“There is no discussion of potential signaling mechanisms between the body wall muscles and the pharynx and how the specifics of ACh signaling might interact with that signaling pathway.”

We are actively seeking to pursue an explanation of this, however, the discussion about why contraction of body wall muscle leads to distal inhibition of pharyngeal pumping in the presence of acetylcholinesterase inhibitor was previously published in Izquierdo PG et al. 2022*.

Based on the reviewer’s comment, the observation and citation to previously published research has been included in the introduction.

*Izquierdo PG, Calahorro F, Thisainathan T, Atkins JH, Haszczyn J, Lewis CJ, Tattersall JEH, Green AC, Holden-Dye L, O'Connor V. Cholinergic signalling at the body wall neuromuscular junction distally inhibits feeding behaviour in Caenorhabditis elegans. J Biol Chem. 2022 Jan;298(1):101466. doi: 10.1016/j.jbc.2021.101466. Epub 2021 Dec 3. PMID: 34864060; PMCID: PMC8801469.

Reviewer 2 Comment 5

Minor - many writing and format flaws were noted which detract from the overall quality and readability of the manuscript. Below is a partial list. Thorough review is needed.

We have shifted the focus of the text and made several directed typological changes. We have proofed the re-submission. 

Text references to figures should indicate which part of the figure is being referenced (i.e. Fig. 7a, Fig. 7b)

Thank you. We have addressed this in the further proof reading of the text.

Line 28 – receptor’s should be receptor

Thank you. We have corrected this.

Line 99- ‘meadiated’ should be ‘mediated’

Thank you. We have corrected this spelling mistake.

Line 105 – ‘incubation to’ should be ‘incubation with’ or ‘exposure to’

Thank you. We have changed to “incubation with”.

Fig. 1 stars indicating significant differences are not shown anywhere on figure

Thank you. We have modified the figure to include statistical significance. Figure 1 and 2 have now been merged according to reviewer recommendation.

Line 143. Should make it clear in the previous sections that aldicarb is a carbamate acetylcholinesterase

Thank you. We have introduced that aldicarb is a carbamate in the sentence above.

Line 151 – should be ‘similar to the aldicarb experiment’

Thank you. We have inserted the definite article in this sentence of the manuscript.

Line 154 – no support is provided for the statement that worms recover pumping 3 hr after removal from paraoxon-ethyl

Apologies for not being specific in the text. The recovery of the pharyngeal function is observed at time 0, being exactly the same pharyngeal pumping value for preconditioned and non-preconditioned worms. Nevertheless, the reference to previous published studies has been added.

178 – grammatical flaw in this heading, unclear meaning

Thank you. We now label this section as “Aldicarb does not precondition a change in paraoxon-ethyl response.”

254 – ‘spontaneous recovery even present’ meaning is unclear

Thank you for highlighting this. The sentence is redundant and has been removed.

255 ‘it’ should be ‘these results’

Thank you. This has been corrected.

258-259 heading is confusing and needs to be reworded. Notably, no molecular imaging is shown, so the location of any protein cannot be stated on the basis of the data

Thanks for this. We have changed this section title to “Body wall muscle L-type receptor function controls drug induced pump inhibition and spontaneous recovery”.

288 – ‘function than’ should be ‘function as’

Thank you. This change has been made.

461-462 meaning is unclear: “shifting….influencing potency and/or efficacy”

This sentence has been simplified and now reads “MOLO-1 is an auxiliary protein involved in the sensitivity of the L-type acetylcholine receptor (Boulin, Rapti et al 2012). Thus, the similarity in responses from receptors encoded by subunits derived from the e193 mutation of unc-29 and the e211 mutation of lev-1 indicates they may have a change in agonist efficacy.

955 – incubated on better than intoxicated onto

Thank you. This change has been incorporated into the new figure legend.

963 – “OP” abbreviation undefined

Thank you. We have spelled the word organophosphate and removed the abbreviation.

964 – incubated on better than incubated onto. Other instances of this flaw also noted (line 1007)

Thank you. This has been corrected in the places indicated and around other sections in the manuscript.

1079 – spontaneous recovery, not ‘spontaneously recovery’

Thank you. This has been changed to reflect the referee’s suggestion.

Fig 7 – which comparison does the uppermost of the 3 bars across the top indicate? The end of the line does not neatly align with either the 6 or the 24 hr time point, but lies in between

Thank you. The upper most line shows that there is a significant difference in the overall pharyngeal response to paraoxon-ethyl exposure between N2 wild type and the rest of the mutants presented in the figure. It does not specifically indicate any of the end-point times represented in the individual graphs.

Line 316 ‘pb’ should be ‘bp’

Thank you. This change has been made.

---

## [Decision Letter · Decision Letter 1]

2 Feb 2023

PONE-D-22-22312R1Modelling organophosphate intoxication in C. elegans highlights nicotinic acetylcholine receptor determinants that mitigate poisoning.PLOS ONE

Dear Dr. O'Connor,

Thank you for submitting your manuscript to PLOS ONE. After careful consideration, we feel that it has merit but does not fully meet PLOS ONE’s publication criteria as it currently stands. Therefore, we invite you to submit a revised version of the manuscript that addresses the points raised during the review process.Overall, the manuscript is considerably improved. However, the reviewers asked for minor changes to further improve your manuscript. As you would not require any experiments, I hope that you could re-submit the revised manuscript very soon.

We look forward to receiving your revised manuscript.

Kind regards,

Hongkyun Kim

Academic Editor

PLOS ONE

Journal Requirements:

Reviewers' comments:

Reviewer's Responses to Questions

**Comments to the Author**

1. If the authors have adequately addressed your comments raised in a previous round of review and you feel that this manuscript is now acceptable for publication, you may indicate that here to bypass the “Comments to the Author” section, enter your conflict of interest statement in the “Confidential to Editor” section, and submit your "Accept" recommendation.

Reviewer #1: (No Response)

Reviewer #2: All comments have been addressed

2. Is the manuscript technically sound, and do the data support the conclusions?

Reviewer #1: Yes

Reviewer #2: Yes

3. Has the statistical analysis been performed appropriately and rigorously? 

Reviewer #1: Yes

Reviewer #2: Yes

4. Have the authors made all data underlying the findings in their manuscript fully available?

Reviewer #1: Yes

Reviewer #2: Yes

5. Is the manuscript presented in an intelligible fashion and written in standard English?

Reviewer #1: Yes

Reviewer #2: Yes

6. Review Comments to the Author

Reviewer #1: Izquierdo et al have made significant improvements to the text, which have greatly strengthened the manuscript. While I find the “n” to be somewhat low in some figures, I believe the data are convincing as the error is small. I appreciate that the authors included their raw data and it is clearly laid out in the accessory document. Below are some remaining minor concerns about the text, which should be easy to address.

• Generation of the lev-1 rescue constructs and strains is not described in the methods

• While supplemental figures are referred to, I couldn’t find them. Perhaps I just missed the link?

• An ANOVA, not a t-test should be used to determine statistical significance in Figure 9

• Figure 9 legend title not accurate

• Figure 11 legend should be reworded as it jumps around from body wall to pharyngeal muscles

• In general, the figure legends are long and repetitive with the manuscript text; in many places could be shortened to make the manuscript more concise

• Line 144-147: Could there be an off target effect of paraoxon-ethyl?

• Line 340: concluding sentence needed

The manuscript still needs additional careful proofreading – here are some suggested changes:

Line 25: Perhaps change “We highlight” to “We show” or “We discovered”

Line 26: Change to “data suggest”

Line 65: Change to “drug-induced”

Line 100: Change from “onto” to “on”

Line 154: Remove bullet point

Line 261,322: Italics needed

Line 281: extra period

Lines 342 and 343: remove the word “of”

Line 344: remove “by the binding”

Line 353: change “was” to “can be” since this is just a model

Line 847: L4+1 is jargon – please reword

Line 863: OP is jargon – please reword

Line 886: Do you mean shrinkage at 1 hour?

Line 891: delete N2 – throughout the paper, use wild type as N2 is jargon

Reviewer #2: Upon re-review of the revised manuscript, I feel my comments have been addressed satisfactorily. However, please replace "REF" on line 21 with a literature citation

7. PLOS authors have the option to publish the peer review history of their article (what does this mean?). If published, this will include your full peer review and any attached files.

Reviewer #1: No

Reviewer #2: No

---

## [Author Response · Author response to Decision Letter 1]

20 Mar 2023

Dear Hongkyun Kim,

We are enclosing the response to reviewers related to the resubmission of our paper entitled “Modelling organophosphate intoxication in C.elegans highlights nicotinic receptor determinants that mitigate poisoning”. This resubmission was originally sent on 9-01-2023. Your editorial direction indicated that the new set of referee responses required what was deemed as modest changes.

We embedded these modest changes below. We show the referees query followed by our considered response listed as they appear in the referees’ reports. We appreciate their efforts and the opportunity for clarification.

Reviewer #1

Generation of the lev-1 rescue constructs and strains is not described in the methods

The referee is correct. The generation of the lev-1 constructs and strains is not described in this paper. These were generated for another publication and the methods. We now indicate this in the method section. This direction is encapsulated in the text: “The transgenic lines VLP1: CB211 lev-1 (e211) IV; Ex[Punc-122::gfp]; VLP2: CB211 lev-1 (e211) IV; Ex[Punc-122::gfp; Pmyo-3::lev-1] were previously available in the laboratory stock [37].” Reference 37 taking the reader to the fuller description of how these reagents were generated.

While supplemental figures are referred to, I couldn’t find them. Perhaps I just missed the link?

These supplementary figures were not uploaded with the original re-submission after the first round of referees’ reports. Apologies for this. This has now been corrected.

An ANOVA, not a t-test should be used to determine statistical significance in Figure 9

We used the t-test to compare differences between specified groups within the context of how individual molecular changes influenced relative to respective control. We have clarified this in the methods section.

- First and second columns: This compares the difference in current amplitude of receptors with all wild type subunits expressed versus receptors where the wild type version of lev-1 has been replaced by the mutated version e211. To investigate if lev-1 mutation has a significant effect.

- First and fourth columns: This compares the difference in current amplitude of receptors with all wild type subunits expressed versus receptors where the wild type version of unc-29 has been replaced by the mutated version e193. To investigate if the unc-29 mutation has a significant effect.

- Second and Third columns: This comparison is made to evaluate the effect in current amplitude of the co-expression of wild type and mutated version of lev-1 in oocytes. To investigate if the mutated lev-1 subunit has potential to compete.

- Fourth and Firth columns: This comparison is made to evaluate the effect in current amplitude of the co-expression of wild type and mutated version of unc-29 in oocytes. To investigate if the mutated unc-29 has potential to compete.

Figure 9 legend title not accurate

Figure legend title has been modified to gain accuracy and now it reads: 

“Figure 9: The mutant genes in lev-1 (e211) and unc-29 (e193) encode for subunits that assemble L-type receptors with reduced function.”

Figure 11 legend should be reworded as it jumps around from body wall to pharyngeal muscles

Thank you for the suggestion. The figure legend has been reworded to add clarity. 

We have added the phrases … this leads to an associated inhibition of pharyngeal function. 

At the level of the body wall acetylcholinesterase…… 

This reinforces the idea that the cartoon is focussed on body wall muscle function that is directly scored by assaying pharyngeal pumping.

In general, the figure legends are long and repetitive with the manuscript text; in many places could be shortened to make the manuscript more concise

We have read through the legends and copy-edited for brevity as long as it does not compromise clarity and our preferred view that figure legends should make the figures as self-contained as possible.

In addition, we removed the text at ln 244 in current submission as it was clearly repetitive and not helpful to the reader. “Compared to wild type nematodes, both lev-1 and ric-3 mutants exhibited a sustained recovery of the pharyngeal function after the spontaneous recovery. Likewise, after the complete inhibition of the pharyngeal function after 3 hours of exposure, ric-3 (hm9) and lev-1 (e211) deficient worms displayed an exaggerated and prolonged within drug recovery of the pharyngeal function compared to the wild type worms (Fig. 6)”.

Line 144-147: Could there be an off target effect of paraoxon-ethyl?

The referee is correct in their assertion and based on the original comments of the referees in the last round of reports we wrote the highlighted sections such that it did not ignore this possibility. To reinforce this we have now added the sentence:

“This might include contributions from non-overlapping targets for the two classes of distinct acetylcholinesterase inhibitors”.

Line 340: concluding sentence needed

Thank you for this prompt. We have now added the sentence “This supports the notion that auxiliary organization, impacting the structure or function of receptor signalling, may underpin these observed changes in the efficacy of organophosphate intoxication.” 

This is specific to this section and the overall concept that we go on to promote in the immediately ensuing discussion.

Line 25: Perhaps change “We highlight” to “We show” or “We discovered”

Highlight has been replaced by identified.

Line 26: Change to “data suggest”

This has been changed to data suggests as suggested by the referee.

Line 65: Change to “drug-induced”

This has been changed in line 65 and everywhere else in the manuscript.

Line 100: Change from “onto” to “on”

This has been corrected.

Line 154: Remove bullet point

Bullet point has been removed.

Line 261,322: Italics needed

Italics has been corrected in the lines suggested and in the rest of the manuscript.

Line 281: extra period

This has been corrected.

Lines 342 and 343: remove the word “of”

These have been removed.

Line 344: remove “by the binding”

Thank you. This has been corrected and now reads ….. the acetylcholine signalling because……

Line 353: change “was” to “can be” since this is just a model

This has been changed.

Line 847: L4+1 is jargon – please reword

This has been removed and the staging informed through methods.

Line 863: OP is jargon – please reword

OP is an accepted abbreviation for organophosphate, however Paraoxon-ethyl has been used to impart precision.

Line 886: Do you mean shrinkage at 1 hour?

Thank you for noticing this. The correction has been made to state “shrinkage at 1 hour”.

Line 891: delete N2 – throughout the paper, use wild type as N2 is jargon

N2 is the name of the strain used as wild type throughout the manuscript. We have added wild type after N2.

Reviewer #2: 

……please replace "REF" on line 21 with a literature citation

We apologies for this mistake. The corresponding reference 62 has been added.

---

## [Decision Letter · Decision Letter 2]

10 Apr 2023

Modelling organophosphate intoxication in C. elegans highlights nicotinic acetylcholine receptor determinants that mitigate poisoning.

PONE-D-22-22312R2

Dear Dr. O'connor

We’re pleased to inform you that your manuscript has been judged scientifically suitable for publication and will be formally accepted for publication once it meets all outstanding technical requirements.

Kind regards,

Hongkyun Kim

Academic Editor

PLOS ONE

Additional Editor Comments (optional):

Reviewers' comments:

Reviewer's Responses to Questions

**Comments to the Author**

1. If the authors have adequately addressed your comments raised in a previous round of review and you feel that this manuscript is now acceptable for publication, you may indicate that here to bypass the “Comments to the Author” section, enter your conflict of interest statement in the “Confidential to Editor” section, and submit your "Accept" recommendation.

Reviewer #1: All comments have been addressed

2. Is the manuscript technically sound, and do the data support the conclusions?

Reviewer #1: Yes

3. Has the statistical analysis been performed appropriately and rigorously? 

Reviewer #1: Yes

4. Have the authors made all data underlying the findings in their manuscript fully available?

Reviewer #1: Yes

5. Is the manuscript presented in an intelligible fashion and written in standard English?

Reviewer #1: Yes

6. Review Comments to the Author

Reviewer #1: (No Response)

7. PLOS authors have the option to publish the peer review history of their article (what does this mean?). If published, this will include your full peer review and any attached files.

Reviewer #1: No

---

## [Editor Report · Acceptance letter]

13 Apr 2023

PONE-D-22-22312R2 

Modelling organophosphate intoxication in C. elegans highlights nicotinic acetylcholine receptor determinants that mitigate poisoning. 

Dear Dr. O'Connor:

I'm pleased to inform you that your manuscript has been deemed suitable for publication in PLOS ONE. Congratulations! Your manuscript is now with our production department. 

Kind regards, 

on behalf of

Dr. Hongkyun Kim 

Academic Editor

PLOS ONE